# Impact of poleward heat and moisture transports on Arctic clouds and climate simulation

Eun-Hyuk Baek[1], Joo-Hong Kim[2], Sungsu Park[3*], Baek-Min Kim[4*], Jee-Hoon Jeong[1]

[1]Faculty of Earth System and Environmental Sciences, Chonman National University, 77 Yongbong-ro, Buk-gu Gwangju, 61186, South Korea

[2]Unit of Arctic Sea–Ice Prediction, Korea Polar Research Institute, 26 Songdomirae-ro, Yeonsu-gu, Incheon, 21990, South Korea

[3]School of Earth and Environmental Sciences, Seoul National University, 1 Gwanak-ro, Gwanak-gu, Seoul, 08826, South Korea

[4]Department of Environmental Atmospheric Sciences, Pukyong National University, 49 Yongso-ro, Nam-gu, Busan, 48513, South Korea

*Correspondence to*: Sungsu Park (sungsup@snu.ac.kr) and Baek-Min Kim (baekmin@pknu.ac.kr)

**Abstract.** Many general circulation models (GCMs) have difficulty simulating Arctic clouds and climate, causing substantial inter-model spread. To address this issue, two Atmospheric Model Inter-comparison Project (AMIP) simulations from the Community Atmosphere Model version 5 (CAM5) and Seoul National University (SNU) Atmosphere Model version 0 (SAM0) with a Unified Convection Scheme (UNICON) are employed to identify an effective mechanism for improving Arctic cloud and climate simulations. Over the Arctic, SAM0 produced a larger cloud fraction and cloud liquid mass than CAM5, reducing the negative Arctic cloud biases in CAM5. The analysis of cloud water condensate rates indicates that this improvement is associated with an enhanced net condensation rate of water vapor into the liquid condensate of Arctic low-level clouds, which in turn is driven by enhanced poleward transports of heat and moisture by the mean meridional circulation and transient eddies. The reduced Arctic cloud biases lead to improved simulations of surface radiation fluxes and near-surface air temperature over the Arctic throughout the year. The association between the enhanced poleward transports of heat and moisture and increase in liquid clouds over the Arctic is also evident not only in both models, but also in the multi-model analysis. Our study demonstrates that enhanced poleward heat and moisture transport in a model can improve simulations of Arctic clouds and climate.

## 1 Introduction

With the increasing amounts of greenhouse gases, the Arctic has undergone the most rapid warming of any location on Earth. During the last decade, the warming rate of the near-surface air temperature over the Arctic has been two to three times that of the entire globe (Johannessen et al., 2016; Screen and Simmonds, 2010; Serreze and Barry, 2011). This pronounced Arctic temperature amplification, some of which is forced by the positive feedbacks among various climate components (e.g., sea ice albedo feedback (Deser et al., 2000), water vapor and cloud feedback (Lu and Cai, 2009), and lapse-rate feedback (Pithan et al., 2014)), is also responsible for extreme weather and climate events over mid-latitude continents (Kug et al., 2015; Screen and Simmonds, 2013; Wu and Smith, 2016). Most general circulation models (GCMs) struggle to simulate the Arctic climate properly, producing results with excessive cold surface temperature. The inter-GCM spread of greenhouse-induced warming is the largest over the Arctic (Boe et al., 2009; de Boer et al., 2012; Chapman and Walsh, 2007; Karlsson and Svensson,

2013). Many researchers have reported that the GCM-simulated cold biases over the Arctic are associated with the shortwave (SW) and longwave (LW) radiation biases at the surface, which are due to poor simulation of Arctic clouds (Barton et al., 2014; English et al., 2015; Karlsson and Svensson, 2013; Shupe and Intrieri, 2004).

Over the Arctic, many GCMs underestimate the cloud fraction (de Boer et al., 2012; Cesana and Chepfer, 2012; English et al., 2015; Kay et al., 2016) and cloud liquid mass (Cesana et al., 2015; English et al., 2014; Kay et al., 2016). Because liquid-containing clouds (i.e., mixed-phase clouds) have larger optical depths than pure ice clouds (King et al., 2004; Shupe and Intrieri, 2004), reduced cloud liquid mass causes weaker cloud radiative forcing in GCMs. Unlike at midlatitudes, mixed-phase clouds over the Arctic can persist for several days (Morrison et al., 2011; Shupe et al., 2011). From a process perspective, cloud liquid in mixed-phase clouds should be rapidly depleted into cloud ice within a few hours owing to the higher saturation vapor pressure over water compared with that over ice (i.e., the Wegener–Bergeron–Findeisen (WBF) mechanism) (Bergeron, 1935; Findeisen, 1938; Wegener, 1911). Therefore, to sustain cloud liquids for several days, a certain production mechanism is necessary to counteract the WBF depletion process. Morrison et al. (2011) reviewed various candidate production processes for cloud liquid in Arctic mixed-phase clouds, such as the compensating feedback between the formation and growth of cloud liquid droplets and ice crystals (Jiang et al., 2000; Prenni et al., 2007), in-cloud turbulence generated by cloud top radiative cooling (Korolev and Field, 2008; Shupe et al., 2008), and horizontal advection by large-scale flows (Sedlar and Tjernström, 2009; Solomon et al., 2011). More recently, researchers have also noted that ice nucleation may be important for correctly simulating Arctic mixed-phase clouds. Liu et al. (2011) demonstrated that their revised ice nucleation scheme increased the cloud liquid mass in Arctic mixed-phase stratocumulus clouds and the associated downward LW flux at the surface during the Fall 2004 Mixed-Phase Arctic Cloud Experiment (MPACE). Subsequent sensitivity studies with various ice nucleation schemes yielded similar results (English et al., 2014; Xie et al., 2013). These improvements are attributed to the revised ice nucleation that decelerates the WBF depletion process in mixed-phase clouds. Even with increased cloud liquid mass, the low-level cloud fraction decreased in the simulations, such that the radiation flux biases at the surface and top of the atmosphere (TOA) still remained.

To determine the factors responsible for the negative biases in GCM-simulated cloud liquid mass and cloud fraction over the Arctic in this study, the Arctic climate simulated by the Seoul National University Atmosphere Model version 0 with a Unified Convection Scheme (SAM0-UNICON; Park, 2014a, 2014b; Park et al., 2017; Park et al., 2019) is compared to that of the Community Atmosphere Model version 5 (CAM5; Neale et al., 2012; Park et al., 2014). By comparing the two Atmospheric Model Intercomparison Project (AMIP) simulations CAM5 and SAM0–UNICON, we elucidate 1) the differences in the cloud properties over the Arctic as simulated by SAM0–UNICON and CAM5, 2) the mechanisms of cloud simulation improvement, and 3) the effects of cloud simulation on the Arctic climate simulation. Section 2 describes the model design and data used in this study. Section 3.1 presents the results of the Arctic cloud simulations and related mechanisms, and Section 3.2 describes the effects of Arctic clouds on the Arctic climate simulations. Finally, Section 4 provides a summary and discussion.

## 2 Method

### 2.1 Model and experimental design

SAM0-UNICON (Park et al., 2019), hereinafter referred to as SAM0 for simplicity, is an international GCM participating in the Coupled Model Intercomparison Project 6 (CMIP6) (Eyring et al., 2016). SAM0 is based on CAM5 but adopts the Unified Convection Scheme (UNICON) (Park, 2014a, 2014b) instead of the shallow (Park and Bretherton, 2009) and deep convection schemes (Zhang and McFarlane, 1995) of CAM5; further, it has a revised treatment of the cloud macrophysics process (Park et al., 2017). Other features, such as the dynamic core, cloud macrophysics and microphysics schemes, planetary boundary layer (PBL) scheme, etc. are exactly the same in both models. UNICON is a process-based subgrid convection parameterization scheme consisting of multiple convective updrafts, convective downdrafts, and subgrid cold pools and mesoscale organized flow without relying on any equilibrium constraints, such as convective available potential energy (CAPE) or convective inhibition (CIN) closures. UNICON simulates all dry–moist, forced–free, and shallow–deep convection within a single framework in a seamless, consistent, and unified manner (Park, 2014a, 2014b). The revised cloud macrophysics scheme diagnoses additional detrained cumulus by assuming a steady-state balance between the detrainment rate of cumulus condensates and the dissipation rate of detrained condensates by entrainment mixing (Park et al., 2017). The addition of detrained cumulus substantially improves the simulation of low-level clouds and the associated cloud radiative forcing in the subtropical trade cumulus regime. Park et al. (2019) showed that the global mean climate, 20th century global warming, and El Niño and Southern Oscillation simulated by SAM0 are roughly similar to those of CAM5 and the Community Earth System Model version 1 (CESM1; Hurrell et al., 2013); however, SAM0 substantially improves the simulations of Madden–Julian Oscillation (MJO) (Madden and Julian, 1971), the diurnal cycle of precipitation, and tropical cyclones, all of which are known to be extremely difficult to simulate in GCMs.

To evaluate the impact of SAM0 on the Arctic cloud system, we conducted five ensemble experiments of an AMIP simulation for 36 years from January 1979 to February 2015 with a horizontal resolution of 1.9° latitude × 2.5° longitude and with 30 vertical layers for both CAM5 and SAM0. The climatology from the two simulations over the Arctic is then compared. The detailed settings of the AMIP simulations is identical to those described in Park et al. (2014). For rational comparison with satellite observation data, the model cloud fraction is calculated using the lidar simulator in the Cloud Feedbacks Model Intercomparison Project (CFMIP) Observation Simulator Package (COSP) diagnostic model. A detailed description of the COSP diagnostic model can be found in Kay et al. (2012).

### 2.2 Observational data

The observed Arctic cloud fraction and condensate phase information are obtained from the Cloud-Aerosol Lidar and Infrared Pathfinder Satellite Observations (CALIPSO)–GCM Oriented CALIPSO Cloud Product (CALIPSO–GOCCP) from June 2006 to November 2010 (Chepfer et al., 2010). The lidar beam of CALIPSO may not detect a few ice crystals underneath optically thick stratocumulus clouds due to attenuation and CALIPSO–GOCCP may slightly underestimate the ice clouds in the lowest levels at midlatitudes and in polar regions (Cesana et al., 2015). Nevertheless, CALIPSO–GOCCP currently provides the best available satellite observations of polar clouds because it can detect optically thin clouds without relying on the albedo or thermal contrast (Cesana and Chepfer, 2012; Kay et al., 2012). The observed TOA fluxes are obtained from version 2.8 of the Clouds and Earth's Radiant

Energy System (Wielicki et al., 1996) Energy Balanced and Filled data (Loeb et al., 2009) (CERES–EBAF) from March 2000 to February 2013. Although CERES–EBAF likely exceeds the global uncertainty particularly for clear sky retrieval over the Arctic due to the low albedo contrast between snow and clouds, it is the only available source of basin-wide TOA fluxes in the Arctic, and newer versions have advanced to distinguish clouds from underlying high-albedo sea ice and snow cover by utilizing cloud radiances from the collocated Moderate Resolution Imaging Spectroradiometer (MODIS) and sea ice concentration fields from the National Snow and Ice Data Center (NSIDC) (English et al., 2014). The climatology data of long-term ground-based cloud and radiation measurements from 1998 to 2010 at the North Slope of Alaska (NSA) Barrow site (71.38° N, 156.68° W) from the Atmospheric Radiation Measurement (ARM) Best Estimate (ARMBE) dataset (Xie et al., 2010) are used for the model evaluation. The Arctic near-surface air temperature at a height of 2 m ($T_{2m}$), liquid water path (LWP), and ice water path (IWP) are obtained from the European Center for Medium-Range Weather Forecasts (ECMWF) ERA-Interim reanalysis dataset from January 1979 to February 2015 (Dee et al., 2011).

## 2.3 CMIP5 models

To identify the relationship between the Arctic clouds and poleward transports of moisture and heat, we also analyzed AMIP simulations of Coupled Model Intercomparison Project Phase 5 (CMIP5) (Taylor et al., 2012). We used the outputs from nine models (bcc-csm1-1-m, CanAM4, CNRM-CM5, GFDL-CM3, HadGEM2-A, IPSL-CM5A-MR, IPSL-CM5B-LR, MIROC5, and MPI-ESM-LR), which can be accessed from http://pcmdi.llnl.gov/. These models are selected based on the availability of the following model outputs: monthly low-cloud fraction calculated using the CALIPSO COSP diagnostic model (variable name: cllcalipso), liquid water path (variable name: clwvi), ice water path (variable name: clivi), daily meridional wind (variable name: va), air temperature (variable name: ta), and specific humidity (variable name: hus).

## 3 Results

### 3.1 Arctic clouds and their relationships with poleward moisture and heat transports

SAM0 reduces the negative biases of CAM5 in cloud fraction and liquid cloud simulations. Figure 1a shows the annual cycle of the total cloud fraction (TCA) averaged over the Arctic area (north of 65° N) obtained from CAM5, SAM0, and observation. Consistent with Kay et al. (2012) and English et al. (2014), CAM5 underestimates the observed TCA throughout the year. The negative biases in the CAM5-simulated TCA are reduced in SAM0, which simulates a more realistic TCA, particularly during summer. SAM0 improves not only the cloud fraction but also the simulation of cloud phase characteristics. Cesana et al. (2015) proposed the height at which the ratio of cloud ice mass to total cloud condensate mass is 90 % (i.e., the phase ratio, PR90) as a useful indicator in assessing the model performance to simulate the cloud phase. For most GCMs, PR90 is lower than the satellite observation height, implying that most GCMs underestimate cloud liquid mass or overestimate cloud ice mass. Both CAM5 and SAM0 underestimate the cloud liquid mass over the Arctic; however, SAM0 yields better estimates than CAM5 (Fig. 1b). Not only the biases against satellite observation, but also the biases against ground-based observation are reduced in SAM0. Figure 2 shows the annual cycles of TCA, LWP, surface downward SW radiation (FSDS), and surface downward LW radiation (FLDS) from CAM5, SAM0, and the observation at the Barrow site. TCA is less in the CAM5 results than in the observation except in July and August.

LWP is also underestimated over the entire period. Accordingly, the downward SW flux is overestimated and the downward LW flux is underestimated, particularly in autumn and winter. Although TCA in SAM0 is overestimated in summer compared with the observation, SAM0 reduces the bias of CAM5 during the other periods. The simulated LWP is closer to the observation than that obtained using CAM5. Biases of the surface radiation fluxes are also reduced, except during summer.

Figure 3 shows the annual-mean vertical profiles of the grid-mean cloud condensate masses and the cloud fraction differences between SAM0 and CAM5 averaged over the Arctic area. Compared with CAM5, SAM0 simulates more cloud liquid condensate mass in the lower troposphere but slightly less cloud ice condensate mass throughout the troposphere (Figs. 3b and 3c). Thus, the total cloud condensate mass increases (decreases) in the lower troposphere (mid-troposphere) from CAM5 to SAM0, respectively, which is responsible for the difference in the cloud fraction (Figs. 3a and 3d). The increase in the cloud liquid condensate mass reduces the bias against the

ERA-interim reanalysis. CAM5 underestimates both the cloud liquid and ice condensation against that in the reanalysis (Supplements S1b and S1e). SAM0, however, simulates cloud liquid condensation close to that in the reanalysis, although the cloud ice condensation is underestimated as much as that obtained using CAM5 (Supplements S1c and S1f). These changes in cloud characteristics from CAM5 to SAM0 differ from those in previous reports on the effects of revised ice nucleation schemes (English et al., 2014; Liu et al., 2011; Morrison

et al., 2008), in which a smaller (larger) low-level (mid-level) cloud fraction was simulated. The increase (decrease) of cloud liquid (ice) mass is consistent with the increase in PR90 height from CAM5 to SAM0, as shown in Fig. 1b.

       To understand the physical processes responsible for the increases in the cloud fraction and cloud liquid mass in

the lower troposphere from CAM5 to SAM0, we plotted the annual-mean vertical profiles of the grid-mean tendencies of cloud liquid and ice condensate masses averaged over the Arctic from various physical processes (Fig. 4). Both CAM5 and SAM0 show two main physical processes generating Arctic cloud liquid condensate: the net condensation of water vapor into cloud liquid (NCD) simulated by the cloud macrophysics scheme and the convective detrainment of cloud liquid (DET). In contrast, two main depletion processes are the precipitation–

sedimentation fallout of cloud condensate (PRS) and WBF conversion of cloud liquid into cloud ice (WBF) simulated by the cloud microphysics scheme. For cloud ice condensate, the main sources are the net deposition of water vapor into cloud ice (NCD), WBF, and convective detrainment of cloud ice (DET), while the main sink is PRS (Fig. 4b). With the exception within the PBL below 950 hPa, the grid-mean tendencies due to subgrid vertical transports of cloud condensates by local symmetric turbulent eddies (PBL) and nonlocal asymmetric

turbulent eddies (CON) are generally smaller than the other tendencies. Near the surface, the PBL scheme operates as a strong source for cloud liquid owing to downward vertical transport of cloud liquid mass from the cloud layers above (Fig. 4a).

       The largest differences between CAM5 and SAM0 are in NCD and DET, particularly for cloud liquid. For cloud liquid, SAM0 simulates a weaker DET but much stronger NCD than CAM5, such that the sum of NCD and DET

simulated by SAM0 is larger than that of CAM5, with a maximum difference of approximately 0.05 g kg$^{-1}$ day$^{-1}$ around 850 hPa, where the differences in the cloud liquid condensate mass and cloud fraction between CAM5 and SAM0 are also maximized (see Fig. 3b). These findings indicate that the increases in the cloud fraction and cloud liquid condensate mass from CAM5 to SAM0 are mainly due to an enhanced NCD for cloud liquid from CAM5 to SAM0. The differences in PBL and CON between CAM5 and SAM0 are relatively small. For cloud ice, the

overall production rate simulated by SAM0 is smaller than that of CAM5, mainly due to the decreases in NCD and DET slightly compensated by the increases in WBF and PRS, which leads to the decrease in cloud ice mass, as shown in Fig. 3c. The SAM0-simulated WBF tendency is slightly larger than that of CAM5 partly due to the larger cloud liquid mass in SAM0. In summary, the increases in cloud liquid mass, cloud fraction, and PR90 from CAM5 to SAM0 shown in Figs. 1 and 3 (which are improvements) are mainly due to the enhanced NCD for cloud

liquid from CAM5 to SAM0. In accordance with the stronger NCD for liquid, the liquid cloud fraction is also increased to satisfy the saturation equilibrium constraint for cloud liquid (see Appendix A of Park et al. (2014)). However, the question regarding what physical process caused the increase in NCD for cloud liquid from CAM5 to SAM0 remained. In both models, the NCD for cloud liquid is explicitly calculated by the saturation equilibrium in the cloud macrophysics scheme, which indicates that a greater NCD for cloud liquid is produced with more

water vapor and lower temperature (Park et al., 2014). Assuming that the Arctic region is a cylinder, the water vapor over the Arctic region can be increased only in two ways: convergence of meridional moisture flux and surface moisture flux. Because the difference in surface moisture flux between the two models is much smaller than the difference in the convergence of meridional moisture flux in the Arctic region (compare Supplement S2a with S2b), it can be inferred that the difference in the large-scale horizontal advection of moisture from sub-Arctic

to Arctic causes the increase in the Arctic water vapor source. Figure 5 shows the differences in the zonal-mean meridional transports of heat and moisture in the high-latitude region and vertical profiles of water vapor (Q), air temperature (T), and relative humidity (RH) averaged over the Arctic area. The zonal-mean meridional flux can be calculated as shown in Eq. (1):

$$[\overline{vX}] = [\overline{v}][\overline{X}] + [\overline{v^*}\overline{X^*}] + [\overline{v'X'}], \tag{1}$$

where X = Q or T; v is the meridional velocity; the overbars and primes denote time-mean and departure from the time-mean, respectively; and the square brackets and asterisks denote zonal-mean and departure from the zonal-mean, respectively. The first term on the right-hand side is the flux due to the mean meridional circulation, the second term is the flux caused by stationary eddies, and the last term is the flux due to transient eddies.

In the midlatitude and subpolar regions, SAM0 simulates poleward transports of heat and moisture more than

CAM5, particularly in the lower troposphere (Figs. 5a and 5e), mainly due to enhanced transports by mean meridional circulation and transient eddies (Figs. 5b, 5c, 5f, and 5g). The difference in poleward moisture (heat) transport between SAM0 and CAM5 is approximately 10 % (15 %) of the climatology, respectively. The enhanced poleward transports of heat and moisture in SAM0 reduces its bias against the ERA-Interim reanalysis compared with CAM5. CAM5 overestimates both the moisture and heat fluxes over the midlatitude region against the

reanalysis but underestimates those on the periphery (around 70° N) of the Arctic circle (Supplement S3). Although the positive bias over the midlatitude region remains, SAM0 reduces the biases of CAM5 on the periphery (around 70° N) of the Arctic circle (Supplement S4). In the northern hemisphere, SAM0 simulates higher pressure and temperature in the low-latitude region but lower pressure and temperature in the high-latitude region compared with CAM5, which reduces the bias of CAM5 (Supplement S5). The circulation change in SAM0

enhances the mean meridional circulation and polar jet stream over higher latitudes (Li and Wang, 2003). The associated strengthening of the zonal mean meridional wind in the midlatitude region (see the contour lines in Figs. 5b and 5f) enhances the poleward transports of heat and moisture near the surface. The enhanced polar jet stream (see the contour lines in Figs. 5c and 5g) strengthens the storm track activity on the periphery of the Arctic circle (between 60° N and 70° N) (Supplements S5c and S5f) and increases the associated poleward transports of

heat and moisture by transient eddies. Moreover, SAM0 simulates the convection more strongly than CAM5, particularly in most of the tropical ocean, which reduces the bias from the reanalysis (Supplement S6). Several previous studies have shown that enhanced convective activity in the tropics enhances the poleward heat and moisture transport by inducing Rossby wave trains from the tropics toward the pole promoting warm and moist advection from midlatitudes to the Arctic (Lee et al., 2014; Fluorny et al., 2015). As with those studies, SAM0

seems to capture Rossby wave trains emanating from the tropics better than CAM5 (Supplement S5c), leading to enhanced poleward heat and moisture transport in SAM0.

    Consequently, SAM0 simulates higher Q, T, and RH than CAM5 over the Arctic (Figs. 5d and 5h). More poleward transport of moisture in SAM0 enhances the NCD for cloud liquid, as shown in Figs. 3b and 4a. Because the liquid cloud fraction is a function of the grid-mean RH in both models, the cloud fraction increases in the lower

troposphere (i.e., below 700 hPa), as shown in Fig. 3d. In addition, warming associated with enhanced poleward heat transport and condensation heating is likely to reduce the amount of cloud ice mass from CAM5 to SAM0, as shown in Fig. 3c, hence reducing the ice cloud fraction in the mid-troposphere (i.e., above 700 hPa) formulated as a function of cloud ice condensate mass in both models (Fig. 2d).

    The relationships between the poleward moisture transport and NCD for cloud liquid are well reflected by the

seasonal and interannual variabilities in both models (Figs. 6 and 7). SAM0 simulates more poleward moisture transport into the Arctic than CAM5 throughout the year (Fig. 6). In both models, the poleward moisture transports at 65° N is the largest from summer to autumn, and the associated NCD for cloud liquid averaged over the Arctic region nearly agrees with the poleward moisture transport. The seasonal variability of the NCD difference for cloud liquid is almost coincident with that of RH, which explains the increase in the Arctic liquid cloud fraction

from May to September, as shown in Fig. 1. The interannual variations of the poleward moisture transport and NCD for cloud liquid in each model are also highly correlated with the correlation coefficients of 0.84 and 0.81 for CAM5 and SAM0, respectively (Figs. 7a and 7b). In addition, in almost all years, SAM0 simulates more poleward moisture flux and higher NCD for cloud liquid over the Arctic than CAM5, and the inter-model differences of these variables are also highly correlated (Fig. 7c). In summary, the strengthened poleward moisture

transport increases NCD for cloud liquid, cloud liquid mass, and cloud fraction from CAM5 to SAM0.

    The close association between the Arctic cloudiness and poleward transports of heat and moisture, as demonstrated by the analysis of the CAM5 and SAM0 simulation results, also exist in other climate models. Figure 8 shows the scatter plots between the annual mean meridional transports of heat and moisture at 65° N and Arctic cloudiness and the LWP ratio (i.e., the ratio of LWP to the total condensate water path, LWP/(LWP+IWP))

obtained from the analysis of various AMIP simulations of CMIP5 models. Wide inter-model spread exists in the TCA, low cloud fraction (LCA, defined as fractional coverage by clouds between the surface and 700 hPa), LWP ratio, and poleward transports of heat and moisture. Except for a few outliers (e.g., bcc-csm1-1-m and MPI-ESM-LR), there is a clear inter-model proportional relationship between the meridional moisture transport and TCA and LCA (Figs. 8a and 8b). All of the models simulate consistently positive poleward moisture transport.

However, some models simulate equatorward heat transport at 65° N, and the corresponding LWP ratio over the Arctic tends to be smaller than those from the models with poleward heat transport (Fig. 8c). The models with strong poleward moisture transport tend to have strong poleward heat transport as well. The inter-model analysis supports our conclusion that poleward moisture and heat transport is one of the key factors controlling the LCA and LWP in the Arctic.

## 3.2 Impact of Arctic clouds on the Arctic climate

Clouds play a critical role in the surface radiative balance as a climate regulator in the Arctic region. Figure 9 shows the biases of LCA, the upward LW radiation flux at the TOA (FLUT), and $T_{2m}$ during winter obtained from CAM5 and SAM0. As shown, CAM5 suffers from negative biases of LCA, FLUT, and $T_{2m}$ during December, January, and February (DJF) (Fig. 9, left panel). In the Arctic during winter, the lower LCA in CAM5 reduces FLUT over the land and sea–ice region in the lower troposphere because the temperature in the cloudy layer is higher than that at the surface (i.e., temperature inversion). The lower LCA also reduces the downward LW radiation at the surface (FLDS), which leads to colder near-surface air than in the reanalysis and thus enhances the temperature inversion. Compared with CAM5, SAM0 simulates greater LCA, FLUT, and $T_{2m}$ over the entire Arctic (Fig. 9, center panel), such that their negative biases in CAM5 are alleviated by SAM0 (Fig. 9, right panel). Over the ocean where temperature inversion does not exist, the greater LCA in SAM0 results in more FLUT than in CAM5 (Fig. 9e). SAM0 also simulates stronger FLDS than CAM5 over the entire Arctic, as expected (not shown).

Not only the biases during DJF, but also the summer biases of TCA, SW cloud radiative forcing at the TOA (SWCF), and $T_{2m}$ are also reduced from CAM5 to SAM0 (Fig. 10). In most Arctic areas except for some portions of the northern continents, CAM5 yielded negative TCA bias during June, July, and August (JJA) (Fig. 10a). SAM0 simulated a greater TCA than CAM5 (Fig. 10b), such that most of the negative TCA biases in CAM5 over the Arctic sea ice and open ocean areas disappear (Fig. 10c). The increase of TCA in SAM0 causes the SWCF to decrease (Fig. 10e), which also reduces positive SWCF bias in the vicinity of the Arctic pole (Fig. 10f). In the Arctic during summer, cloudiness has the opposite effect on SWCF and LWCF (Supplement S7); thus, it is necessary to examine the net forcing of SW and LW radiation at the surface to determine the impact of Arctic clouds on the Arctic climate. With greater TCA than CAM5, SAM0 simulates more net LW radiation at the surface (FLNS, Fig. 11b). Owing to the high albedo of the underlying sea ice and snow in the vicinity of the Arctic pole, the net SW radiation at the surface (FSNS) does not change much there; however, FSNS decreases substantially in the regions surrounding the Arctic pole (Fig. 11a). Overall, the increase in FLNS dominates over the decrease in FSNS at the Arctic pole, while the opposite is true in the surrounding regions (Figs. 11b and 11c). The net forcing of SW and LW radiation at the surface causes $T_{2m}$ to increase at the Arctic pole and decrease in the surrounding continental area from CAM5 to SAM0 (Fig. 10h). It is possible that the associated increase of $T_{2m}$ from CAM5 to SAM0 in the Arctic pole (Fig. 10h) decreases snow depth, but other factors (e.g., less snowfall) may also be responsible for this decrease (Fig. 11d). Less snow depth results in lower surface albedo (Fig. 11e). The enhanced SWCF cooling near the Arctic pole in SAM0 (Fig. 10e) is the combined results of the increased TCA and decreased surface albedo. If the Arctic sea ice fraction is allowed to change in response to the changes in the overlying atmospheric conditions (e.g., coupled simulation), SAM0 is likely to simulate a lower sea ice fraction than CAM5 due to the increased TCA and warmer near-surface air temperature, which can be further accelerated by the positive surface albedo feedback (Holland and Bitz, 2003). In fact, Park et al. (2019) found that SAM0 simulates lower sea ice fraction than CESM1 (coupled model of CAM5; Hurrell et al., 2013) over the Arctic in a 20th century coupled simulation.

## 4. Summary and Discussion

Many GCMs suffer from cold bias over the Arctic, which has been speculated to be caused by radiation biases associated with cloud fraction and cloud liquid mass underestimation over the Arctic. To address this issue, we compared various aspects of the Arctic clouds and climate in two different AMIP simulations generated by CAM5 and SAM0.

Similar to other GCMs and previous studies, CAM5 underestimates the cloud fraction and cloud liquid mass in the Arctic lower troposphere throughout the year. SAM0 alleviates these problems, although biases still persist. Our analysis showed that this improvement in the Arctic cloud simulation with SAM0 is mainly due to the stronger NCD for cloud liquid, which in turn, was due to enhanced poleward transports of heat and moisture by the mean meridional circulation and transient eddies. In SAM0, UNICON strengthens and shifts poleward the zonal mean meridional circulation, polar jet stream, and associated synoptic storm activity on the periphery of the Arctic circle. The proportional relationship between the Arctic cloudiness and meridional transports of heat and moisture exists not only in the CAM5 and SAM0 models, but also in CMIP5 models. Due to the deficient simulations of cloud fraction and cloud liquid mass, CAM5 suffers from negative near-surface air temperature bias throughout the year. With a greater cloud fraction and cloud liquid mass, SAM0 also alleviates the cold temperature biases in the Arctic mainly by enhancing the downward LW radiation at the surface, which is consistent with the hypotheses suggested by previous researchers (Barton et al., 2014; Chan and Comiso, 2013; Klocke et al., 2011; Pithan and Mauritsen, 2014; Walsh and Chapman, 1998). Our study indicates that enhanced poleward heat and moisture transport can improve simulations of Arctic clouds and climate.

**Author Contributions**

E.-H. Baek performed the overall numerical experiments and analysis. S. Park developed and provided SAM0 and CAM5 and helped analyze the simulation results. B.-M. Kim designed the project and helped analyze the simulation results and the CMIP5 models. All authors contributed to the analyses.

**Competing interests**

The authors declare that they have no conflicts of interest.

**Acknowledgments**

This work was supported by the Korea Polar Research Institute projects entitled "Earth System Model-based Korea Polar Prediction System (KPOPS-Earth) Development and Its Application to the High-impact Weather Events originated from the Changing Arctic Ocean and Sea Ice (PE20090)." E.-H. Baek and J.-H. Jeong are supported by the project entitled "Korea-Arctic Ocean Observing System (K-AOOS), KOPRI, 20160245," funded by the MOF, Korea. S. Park is supported by Seoul National University (SNU). B.-M. Kim is supported by the Korea Meteorological Administration Research and Development Program (grant number KMI2018-03810)

**Data Availability**

The data used in this paper are available at http://gofile.me/6DN3Q/UqIw9dC4i/pub/Paper_data/acp2020/.

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

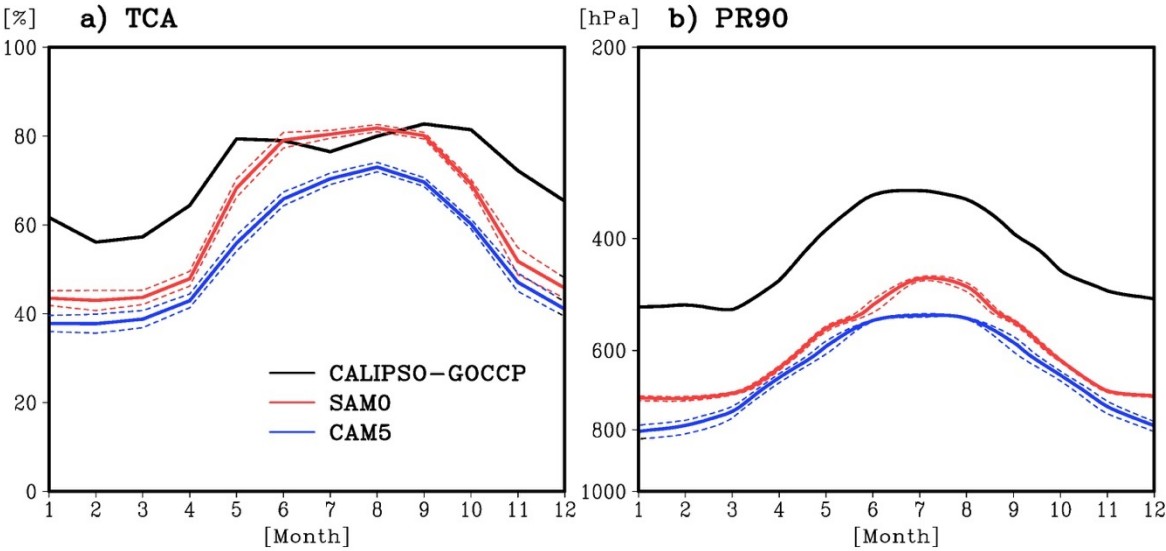

**Figure 1: Annual cycles of the (a) total cloud fraction (TCA) and (b) height at which the ratio of ice condensate mass to total condensate mass is 90 % (phase ratio, PR90) averaged over the Arctic area, north of 65° N from CALIPSO-GOCCP observations averaged from June 2006 to November 2010 (black line), SAM0 (red line), and CAM5 (blue line). The dashed lines denote the standard deviations of the variables. The CALIPSO-GOCCP observations were averaged from June 2006 to November 2010, and the model results are the means of AMIP simulation results for 36 years from January 1979 to February 2015.**

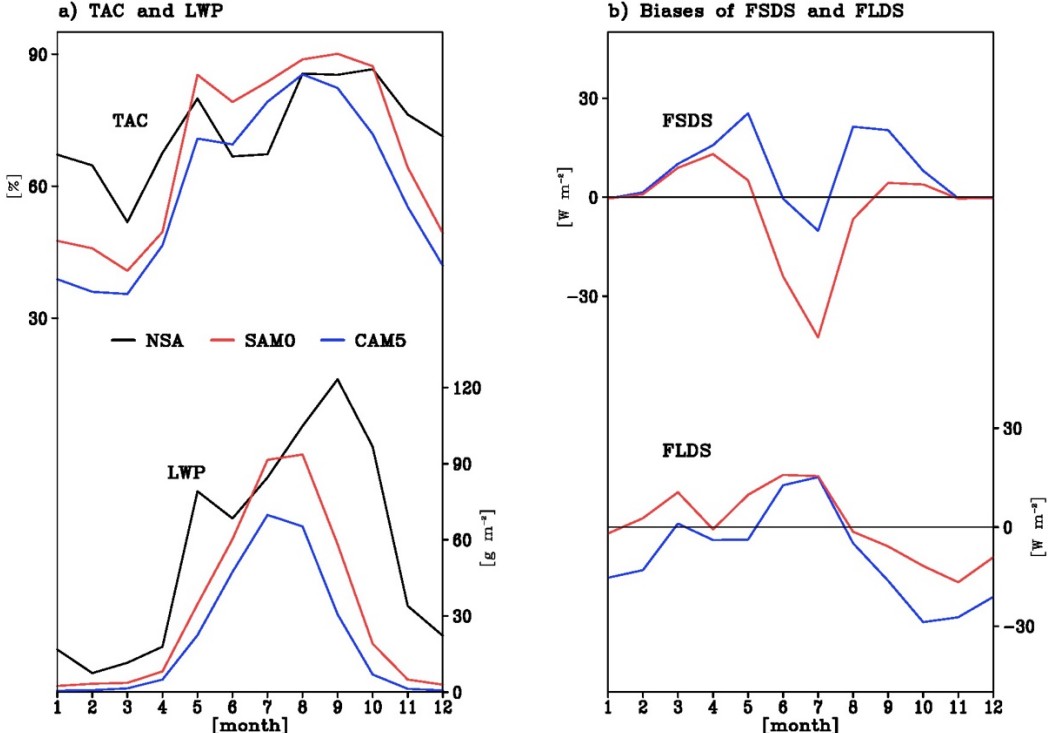

**Figure 2: Annual cycles of total cloud fraction (TAC, upper in (a)) and liquid water path (LWP, bottom in (a)) from the climatology of ground-based cloud and radiation measurements at the North Slope of Alaska (NSA) Barrow site (black line), SAM0 (red line), and CAM5 (blue line). Biases of surface downward SW flux (FSDS, upper in (b)) and surface downward longwave flux (FLDS, bottom in (b)) of SAM0 (red line) and CAM5 (blue line) against the**

climatology of ground-based cloud and radiation measurements at NSA Barrow site. The NSA Barrow site observations were averaged from 1998 to 2010, and the model results are the means of AMIP simulation results for 36 years from January 1979 to February 2015.

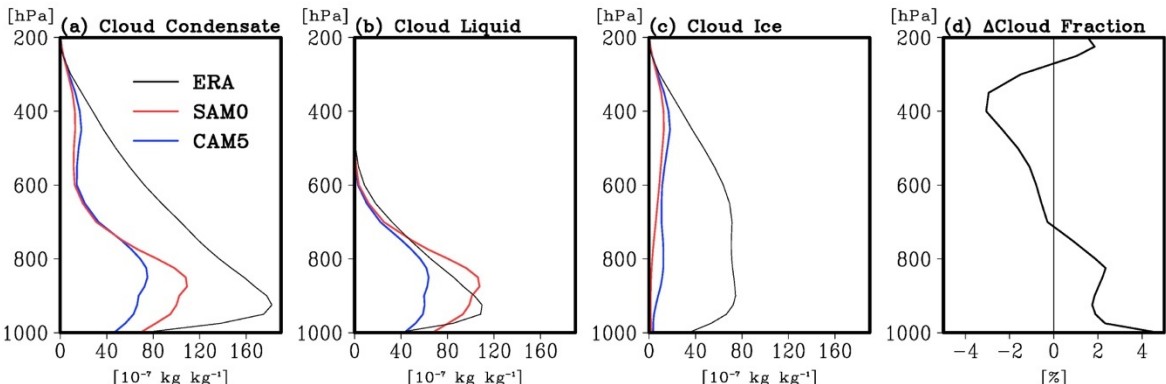

**Figure 3: Annual-mean vertical profiles of grid-mean (a) cloud condensate mass (cloud liquid + cloud ice), (b) cloud liquid mass, and (c) cloud ice mass averaged over the Arctic from ERA-Interim reanalysis (ERA, black lines), SAM0 (red lines) and CAM5 (blue lines), and (d) the difference in cloud fraction between SAM0 and CAM5. ERA-Interim reanalysis was averaged from January 1979 to February 2015, and the model results are the means of AMIP simulation results for 36 years from January 1979 to February 2015.**

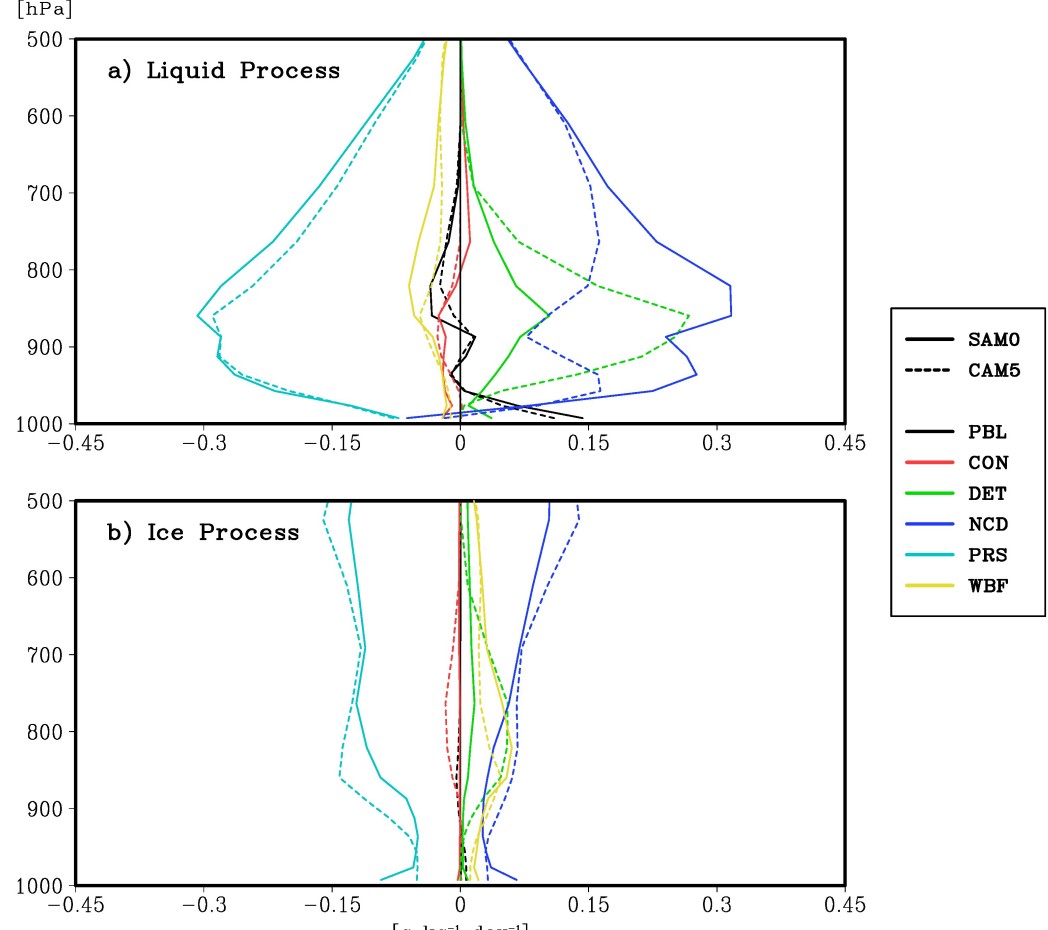



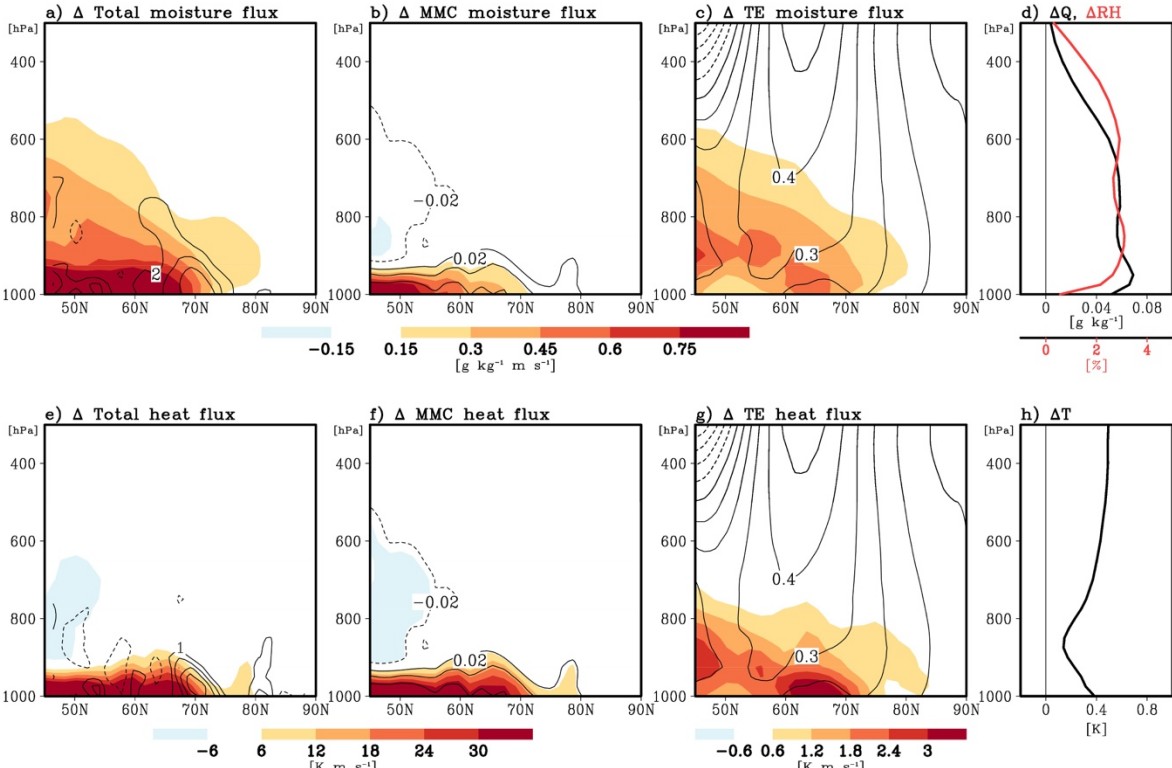

**Figure 4: Annual-mean vertical profiles of the grid-mean tendencies of the (a) cloud liquid mass and (b) cloud ice mass induced by various moist physics processes from SAM0 (solid lines) and CAM5 (dotted lines). The processes shown are subgrid vertical transport by local symmetric turbulent eddies (PBL, black), subgrid vertical transport by nonlocal asymmetric turbulent eddies (CON, red), convective detrainment (DET, green), net condensation of water vapor into cloud liquid and net deposition of water vapor into cloud liquid and ice (NCD, blue), precipitation-sedimentation fallout (PRS, cyan), and WBF conversion from cloud liquid mass to cloud ice mass (WBF, yellow).**

**Figure 5: Differences in zonal-mean meridional fluxes of (a, b, and c) moisture and (e, f, and g) heat due to (a and e) total processes (i.e., the transported sum by mean meridional circulation, stationary eddies, and transient eddies), (b and f) mean meridional circulation (MMC), and (c and g) transient eddies (TE) between SAM0 and CAM5. Differences in the annual-mean vertical profiles (d) water vapor (Q, black) and relative humidity (RH, red), and (h) air temperature (T) averaged over the Arctic between SAM0 and CAM5. The black lines in (a) and (e) denote the differences in zonal-mean convergence of total moisture flux in $10^{-7}$ g kg$^{-1}$ m s$^{-1}$ and total heat flux in $10^{-5}$ K s$^{-1}$. The black lines in (b) and (f) denote the differences in zonal mean meridional wind in m s$^{-1}$. The black lines in (c) and (g) denote the differences in zonal-mean zonal wind in m s$^{-1}$ between SAM0 and CAM5, respectively. Most shaded areas exceed 95 % significance level from the Student t-test.**



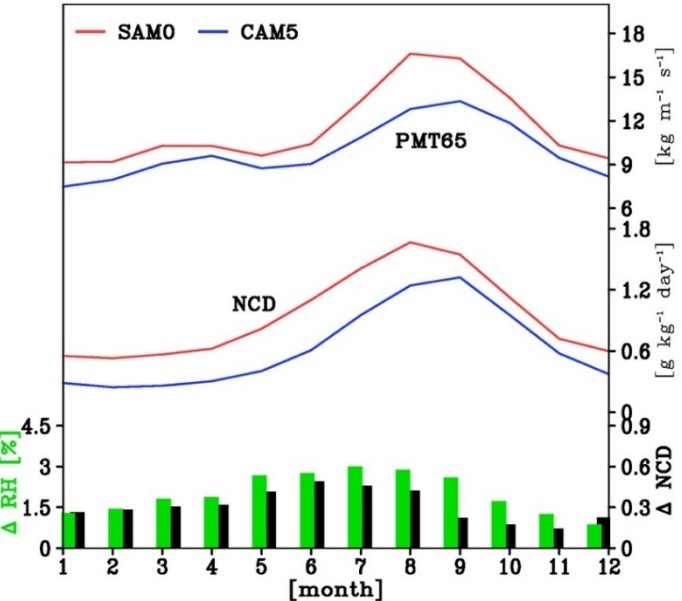

Figure 6: Annual cycles of vertically integrated zonal-mean poleward moisture transport in g kg⁻¹ m s⁻¹ at 65° N (PMT65) and net condensation rate of water vapor into cloud liquid (NCD) in g kg⁻¹ day⁻¹ averaged over the Arctic from SAM0 (red line) and CAM5 (blue line).

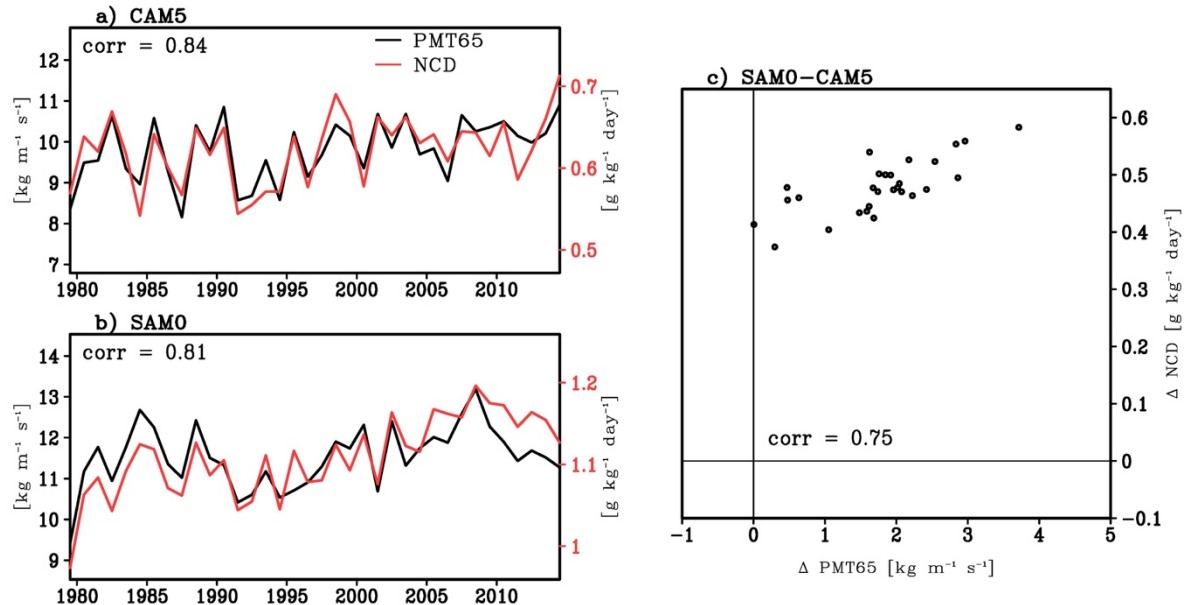

Figure 7: Interannual time series of the vertically integrated annual-mean poleward moisture flux at 65° N (PMT65, black line) and net condensation rate of water vapor into cloud liquid (NCD, red line) averaged over the Arctic from (a) CAM5 and (b) SAM0, and (c) scatter plot of the differences in annual-mean PMT65 and NCD between SAM0 and CAM5.

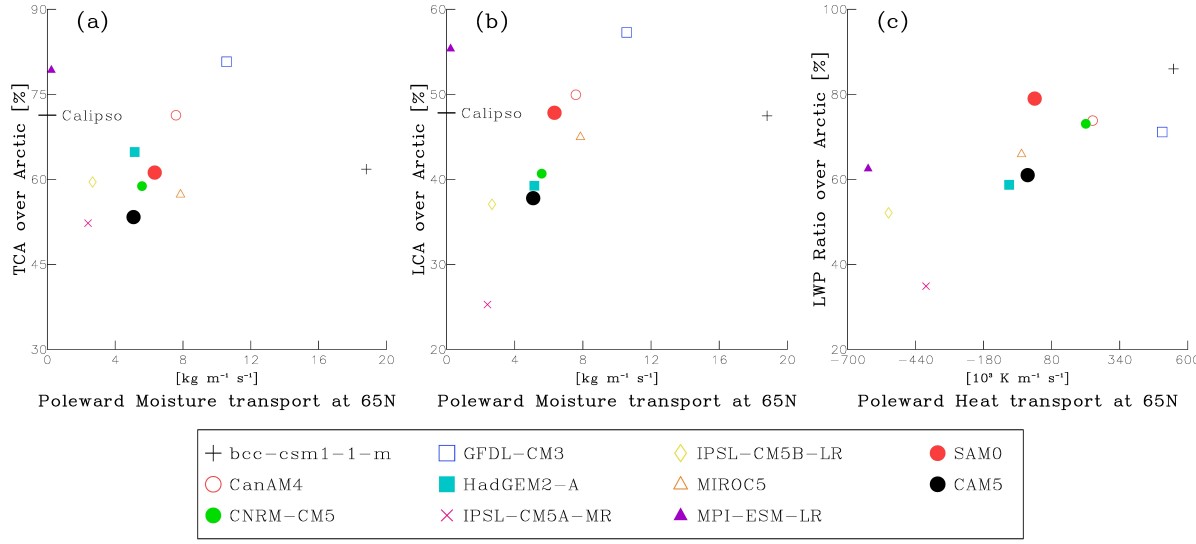

**Figure 8: Scatter plots of the annual mean poleward fluxes of moisture and heat integrated over the vertical layers (1000–7000 hPa) at 65° N, cloud fractions, and LWP ratio averaged over the Arctic, obtained from various AMIP simulations of CMIP5 models, CAM5, and SAM0. The black lines in (a) and (b) denote the observed TCA and LCA, respectively, obtained from CALIPSO-GOCCP data.**


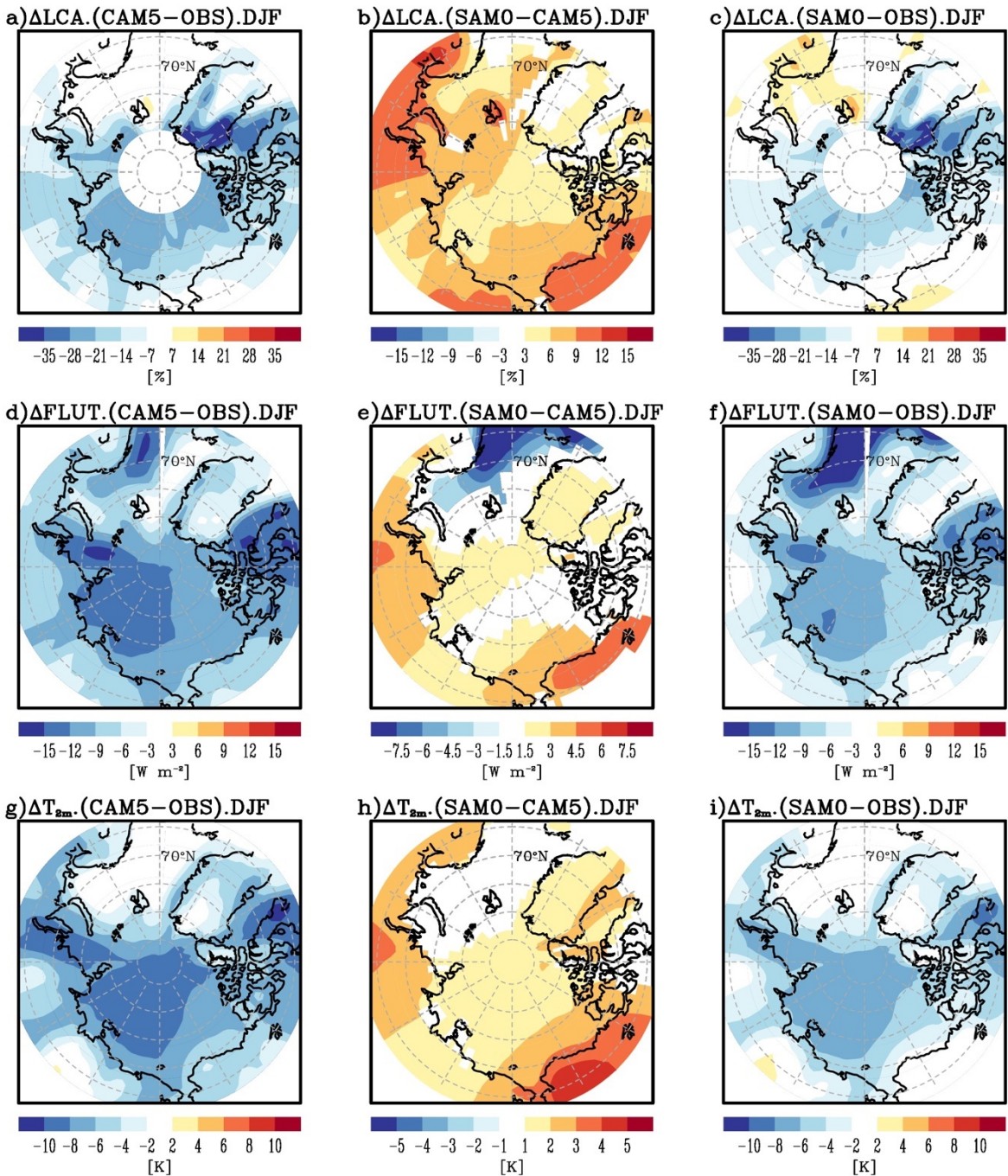


**Figure 9:** Biases of (upper) low cloud fraction (LCA) against the CALIPSO–GOCCP observation, (middle) upward longwave (LW) radiative flux at TOA (FLUT) against the CERES–EBAF observation, and (lower) near-surface air temperature at a 2 m height ($T_{2m}$) against the ERA-Interim reanalysis during DJF obtained from (left) CAM5 and (right) SAM0, and (center) the differences in each variable between SAM0 and CAM5. The CERES-EBAF observations were averaged from 2000 to 2013, ERA-Interim reanalysis was averaged from January 1979 to February 2015, and the model results are the means of AMIP simulation results for 36 years from January 1979 to February 2015. Shaded areas in (b), (e), and (h) exceed 95 % significance level from the Student t-test.


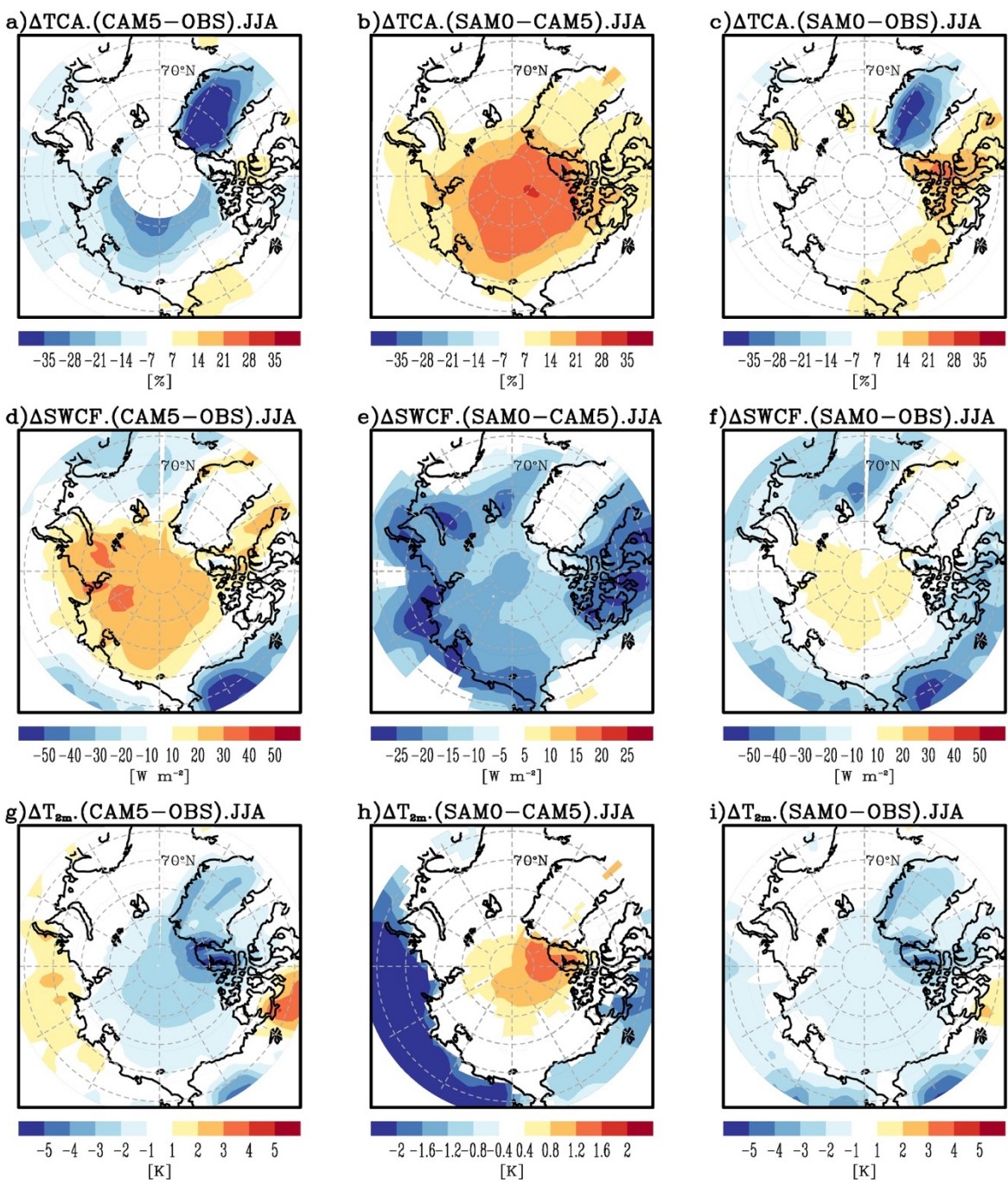

**Figure 10: Identical with Fig. 8, except for total cloud fraction (TCA) in the upper panel, the SW cloud radiative forcing at TOA (SWCF) in the middle panel, and during JJA.**

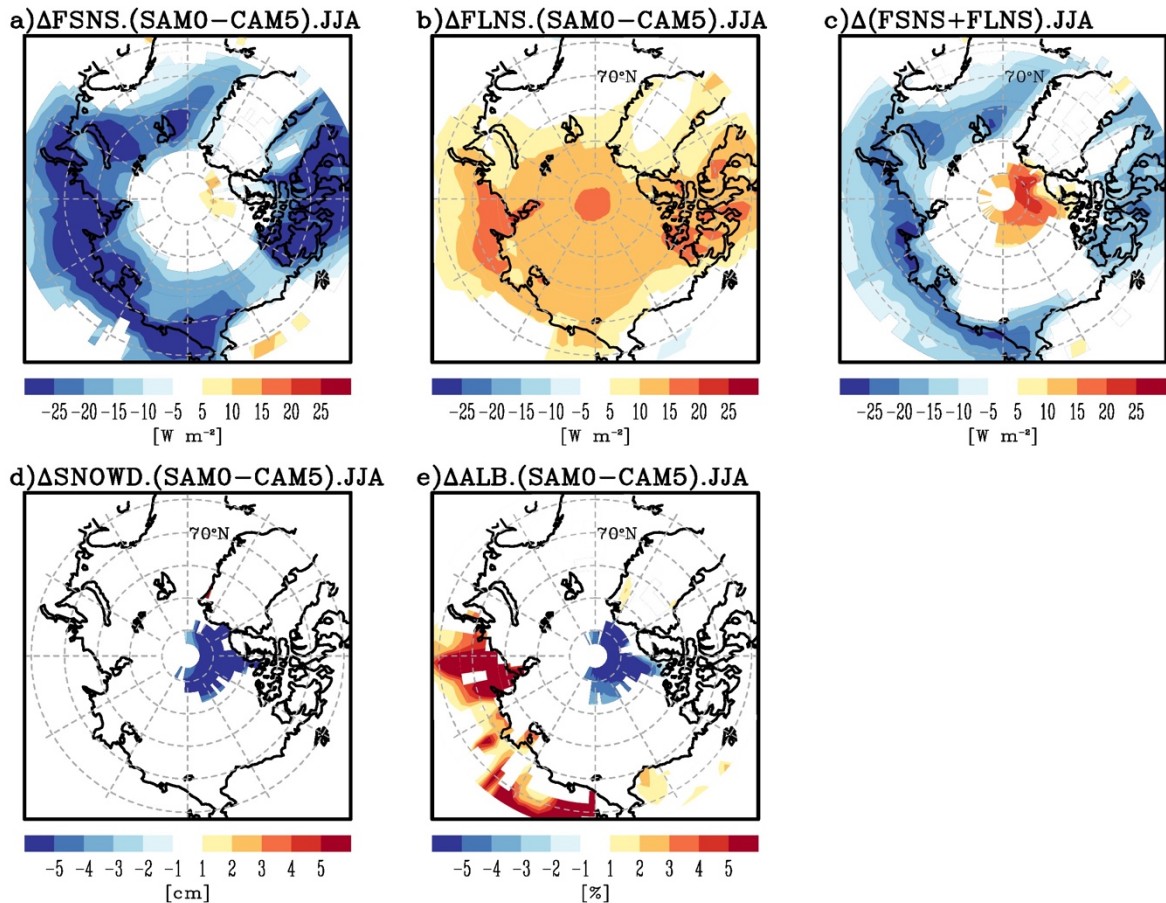

**Figure 11: Differences of (a) net SW flux at the surface (FSNS), (b) net LW flux at the surface (FLNS), (c) sum of FSNS and FLNS, (d) snow depth (SNOWD), and (e) surface albedo (ALB) during JJA between SAM0 and CAM5. Shaded areas exceed 95 % significance level from the Student t-test.**