# Peer review of "Impact of poleward heat and moisture transports on Arctic clouds and climate simulation"

_Atmospheric Chemistry and Physics, 2019_

## Referee Comment (RC1) · Anonymous Referee #2 · 23 May 2019

This manuscript demonstrates the better simulated Arctic clouds and climate in SAM0 than in CAM5 and attributes this improvement to the enhanced poleward transport of heat and moisture. The overall flow is clear and the writing is easy to follow, despite some redundancy and lack of topic sentences. My main concerns are about the logical chain, summarized as follows:

[Figure]

From statement 1 to 3, the arguments are strong. However, from 3 to 5, the evidence is mostly circumstantial. Without corroborating these causal links by more analysis and/or experiments, conclusions of this manuscript could not hold. Please see my detailed comments and suggestions below.

Major issues:

1. For the link between statement 3 and 4, the authors cite Park et al. (2014): "the horizontal and vertical transports of heat and moisture are the important factors inducing the net condensation of water vapor into cloud liquid (NCD) both in SAM0 and CAM5". Are these factors also important in the Arctic regions? How dominant are these factors? Are there any other important factors? Could other modifications in SAM0 contribute to the enhancement of NCD? For example, in-cloud turbulence and precipitation from super-cooled liquid clouds.
   The following analysis of concurrence and correlations between advection and clouds are not definite evidence either. As a model study (instead of observational data analysis), more concrete evidence is expected for this imperative link. How about adding a budget analysis like the one for statement 2 and 3 or experiments turning on/off certain model processes?

2. The authors speculate that horizontal advection rather than vertical transport is responsible for the enhancement of NCD due to the identical surface boundaries. Could the authors provide some discussion instead of speculation on the role of vertical transport? Is it possible that the modification of SAM0 on the convection scheme alters the vertical

structure of the atmosphere, which could also contribute to the NCD enhancement?

3. Evaluated against ERA-Interim, SAM0 exhibits smaller biases than CAM5 in annual-mean geopotential heights simulations. However, this does not guarantee a more accurate simulation of heat and moisture advection. For example, SAM0 overestimates geopotential height at 850 hPa in tropical Pacific and underestimates in sub-Arctic Pacific, leading to larger heat transport into the Arctic. The moisture transport is also affected by the location of pathways. Could the authors evaluate the heat and moisture advection directly against observations?
   More importantly, this is the only way to validate the claim of "**proper** heat and moisture transport is the key process in simulating Arctic climate". Otherwise, the evidence could only support a claim of "enhanced heat and moisture transport improve Arctic climate simulations" because the larger advection could be an overcompensation in order to increase liquid clouds.

4. Page 1 Line 23-26, "proper simulation of poleward heat and moisture transport is one key factor for simulating Arctic clouds" could not be wrong. However, drawing this conclusion from an uncertainty level of "association" hurts rather than lends credibility, especially in a model study. Would the authors provide more evidence to substantiate this claim? Or this manuscript could focus on the improvement of SAM0 and thoroughly evaluate all the causes.

Minor issues:
5. Page 1 Line 9: "with a large inter-model spread" → "causing a large inter-model spread"

6. In the abstract, there needs to be a transition (e.g., your specific question) from the first and second sentences (the big-picture question) to the third (your approaches). Otherwise, readers might wonder how a comparison between two models could possibly address such a complex issue.

7. Page 1 Line 17-18, ""

8. Page 1 Line 18-19, "reducing the Arctic clouds biases in CAM5" → "reducing the **negative** Arctic clouds biases in CAM5"

9. Page 1 Line 19, "budget analysis" is not a common term to me. I suggest removing it from the abstract or at least point it out in Paragraph 2 Page 5 before starting this analysis.

10. Page 1 Line 20, in the rest of this manuscript, "clouds" are used instead of "stratus". Please keep it consistent.

11. Page 2 Line 30-35, it's better to show how the manuscript is organized here rather than presenting results without any analysis.

12. Page 2 Line 34, the phrase "It is/was/will be shown/found" does not convey any messages. It's safe to remove them in most cases to make the writing more concise.

13. Page 4 Figure 1, could the authors add error bars to this figure? e.g., a standard deviation of inter-annual variability.

14. Page 6 Figure 3, have the authors looked at the seasonal breakdown? In winter, could larger moisture transport lead to more NCD to cloud ice?

15. Page 8 Figure 4, what is the percentage of these differences relative to the absolute values of CAM5. If it's too messy to plot on the figure, a description in the text would be fine too.

16. Page 8 Line 12, "" since you've already used "Because".

17. Page 9 Figure 5 a and b, the correlation seems to weaken in the recent years. Any reasons?

18. Page 9 Line 2, ""

19. Page 9 Line 15-16, "All models simulate consistent poleward moisture transport". "Consistent" is not clear. Consistent how?  Suggestions: "All models simulate consistently positive poleward moisture transport" or simply "All models simulate positive poleward moisture transport"

20. Page 10 Figure 6c, I am confused about the unit: $10^3$ K kg m$^{-1}$ s$^{-1}$. Could it be $10^3$ K m$^{-1}$ s$^{-1}$?

21. Page 14 Line 15, ""

22. Most of the paragraphs lack a topic sentence to guide readers. For example, the topic sentence for the first paragraph in Section 3 could be "SAM0 reduces the negative biases of CAM5 in liquid cloud simulations".

---

## Referee Comment (RC2) · Anonymous Referee #3 · 23 May 2019

The paper by Baek et al. analyzes AGCM simulations from two climate models with different convective parameterizations. They show improvements of simulated Arctic cloudiness in SAM0, which they suggest is due to increased moisture transport into the Arctic compared to CAM5. The paper is well written, however, some arguments need to be strengthened and improved. I hope to see the following issues being addressed by the authors.

Major issues:
1) Improvements of simulated Arctic clouds: according to Figure 1, although SAM0 cloud fraction is closer to observations, significant biases still persist, especially in winter. The improvement in PR90 is marginal. I suggest the authors to also compare the liquid and ice water path to the observations (e.g. Lenaerts et al. 2017), because they are also important for cloud radiative effects. For example, does the decrease in cloud ice mass (Figure 2c) make it closer to observations?
2) Relationship between meridional fluxes and increased cloud liquid:
   a) Is vertical advection included in the meridional transport? Heat and moisture transport into the Arctic doesn't just happen in the horizontal plane. In fact, eddies transport moisture along (moist) isentropes.
   b) Even though increased moisture flux and Arctic liquid cloud are correlated, it doesn't provide causation.
3) The results shown here are from atmospheric only GCMs. Since the results depend on the atmospheric heat transport, ocean coupling could potentially alter the results. Have the authors looked at whether the changes in heat and moisture fluxes are still robust in coupled SMA0?

I also agree with Reviewer #2's major issues, some of which are in line with my concerns regarding the causal arguments.

Minor issues:
Please hatch the maps to show their significance level, instead of saying "most shaded areas exceed a 95% significance level". Since not all areas are significant, it is useful to know where it is not significant.

Line-by-line comments:

P3 L7: What microphysics scheme does SAM0 use? Is it the same for CAM5? If not, it can introduce additional sensitivity.

P3 L23: Are different periods from 1979 to 2015 selected to compare with the corresponding observations (CALIPSO and CERES)? If so, it should be specified, since

the simulation period is much longer than what the observations cover. If not, are the results sensitive to the mismatch in periods?

P4 L14-15: The TCA bias also varies seasonally. SAM0's improvement is the most significant in summer, but less so in winter. Have the authors investigated the seasonal and spatial variability in poleward moisture transport? The seasonal cycle combined with spatial maps could shed light on why clouds are underestimated in the Arctic.

P6 Figure 3: Are these budgets closed? Net tendency profiles for liquid and ice from each model can be added to these figures.

P8 Figure 4: According to the moisture and heat flux convergence, we expect increased liquid condensation thus more liquid clouds at around 70N. Is this the case? For example, in winter the total cloud increase is quite spatially uniform over the Arctic Ocean (Fig 7b).

P8 L22-P9 L2: We all know that correlation does not mean causation. By just showing correlation, the causality is not proven. For example, it is also possible that both changes are caused by a third factor that is not in the analysis.

P8 L13: LCA was never spelled out in the paper.

P9 L14: What makes these two models the outliers? Are there any physical reasoning to say so? One should not just pick and choose the models or discard the end members because they do not agree with your hypothesis.

P10 Figure 6: Are the widths of black lines in (a) and (b) represent the spread of observed poleward moisture transport? If not, these lines are very misleading and unnecessary.

P10 L7: Upward LW at TOA includes both clear sky and cloud effects. Have the authors look at the cloud radiative effect differences between the two models? The negative bias in upward LW at TOA in the Atlantic sector seems to get worse in SAM0. Is it because in this region, SAM0 is producing too much clouds comparing to the observations?

P10 L11: Figure 7 shows TCA, but the argument the authors give here involves LCA. Is the LCA change the same as TCA?

P11 L7: Remove "a" before "summertime biases"

P13 Figure 8: Does panel d) suggest that CAM5 SW cloud forcing is too weak at the surface? This would lead to a warm bias over the Arctic ocean. But g) shows a cold bias, which means that the LW cloud forcing bias (not enough warming at surface) dominates the net forcing. It would be helpful to see both LWCF and SWCF to get a fuller picture. I suggest the authors to plot LWCF and SWCF for both TOA and surface, at least to include them in the supplementary material.

P14 L11: The wording "SAM0 remedies these problems" is too strong, given that significant cloud biases still persist.

Reference:
Lenaerts et al. (2017) Polar clouds and radiation in satellite observations, reanalyses, and climate models. GRL.

---

## Referee Comment (RC3) · Anonymous Referee #1 · 13 Jun 2019

This manuscript evaluates the impact of the UNICON unified convection scheme on Arctic cloud biases by comparing AMIP simulations using CAM5 (which implements the Park & Bretherton (2009) and Zhang & McFarlane (1995) convection schemes, against AMIP simulations using the SAM0 model. The authors find that the UNICON scheme increases moisture transport from lower latitudes to the Arctic and results in enhanced cloud fraction and decreases in surface flux and cold surface temperature biases in the Arctic. They show that enhanced poleward moisture transport is also associated with enhanced total and low-cloud cover in the CMIP5 models.

The manuscript presents the potentially interesting finding that poleward moisture and heat transports can somewhat improve the low cloud fraction and liquid biases and therefore radiation budget, surface temperature and sea ice in the Arctic in cli-

mate models relative to observations, but I find that the conclusions drawn from the manuscript are very speculative and based on correlation rather than drawn through rigorous analysis. Major and minor comments follow.

Major comments: * It is not clear whether the only difference between SAM0 and CAM5 is the UNICON scheme. Is this the case? Even so, the scheme itself seems to contain many different changes, making it difficult to isolate cause and effect. There is an awful lot of speculation in the manuscript, from attributing increases in cloud fraction and liquid in the Arctic to increased condensation rate to attributing the increase in condensation rate to poleward transport of heat and moisture. Because it's not clear what exactly is different in SAM0 compared to CAM5 without reading the references in the manuscript in detail, I recommend that the authors conduct sensitivity tests to isolate the individual effects they are speculating. For example, if the authors claim that poleward moisture and heat transports are the main factors in SAM0 that cause an increase in condensation rate in the Arctic, then they could do sensitivity tests where they increase and decrease poleward moisture and heat transports in SAM0 by varying degrees to get a sense of whether or not they are dominant factors in affecting Arctic condensation rate. * The errors of the two observational datasets and Reanalysis data used are not discussed or addressed whatsoever. Please include a detailed description of the errors and biases in all three datasets. In particular, the GOCCP dataset does not account for lidar beam attenuation, which is particularly problematic in the Arctic, where optically thick supercooled liquid clouds attenuate the beam. Precipitating ice particles underneath these layers, which are known to commonly exist, would not be detected. If comparing the results of the models to GOCCP alone in terms of cloud amount, GOCCP might underestimate the actual cloud amount. I suggest that the authors either include a ground-based observational dataset to get an idea of the potential biases involved when comparing the models to GOCCP. * Although SAM0 is able to produce more low cloud amount and cloud liquid and less cloud ice as illustrated in Figure 2, it is not clear from the figures until Figures 6-9 how the models compare against observations. It could be that SAM0 overshoots low-cloud amount/cloud liquid

or undershoots cloud ice relative to the observations. I would suggest including observations in Figure 2 as well. This could be done if the authors were to e.g. run the model in single column mode and compare their results with ground-based observations from the M-PACE field campaign. This could also provide additional evidence to support the authors' claims using an additional complementary ground-based observational dataset. This should also be clarified on lines 17-19 in the Abstract, where it should be specified what observational dataset the reduced biases are with respect to. * Figure 3: Why these microphysical tendencies? Why not include other microphysical processes such as accretion, autoconversion, wet/dry deposition as well? This analysis may be missing processes that are more important than net condensation rate. Also, the nonlinear interactions between the model tendencies are not quantified in Figure 3; the various processes all feed back and are dependent on one another. The authors could make this analysis more rigorous by quantifying the contribution of these liquid and ice tendencies to cloud liquid and ice mass using a multiple linear regression approach. * The strong negative bias in TCA seems to persist in the summer (Figure 8), yet why does there appear to be little to no SWCF bias? Is LCA more relevant than TCA?

Minor comments: Page 9: LCA was never explicitly defined. I'm assuming this stands for low-cloud amount. Does this include clouds from 700hPa to 1000 hPa? * Section 2.1: What is the vertical resolution of the model? What was used as the spin-up time of the model? Which COSP simulator was used (e.g. was it the lidar simulator?) * Figure 3: Does the WBF process include snow? * Section 3.2: The sentences referring to the temperature inversions are written in a confusing way. It seems like the temperature inversions should be mentioned after the effect of LCA on surface fluxes, not before since it's a consequence of the clouds. * Rather than state that "most of the shaded area" is statistically significant, why not shade the statistically insignificant areas to avoid crowding the plot?
* * *
[Figure]

2019.

---

## Author Comment (AC1) · 30 Sep 2019

Response to Reviewer #1

We sincerely appreciate Reviewer #1 for spending his/her invaluable time to give us lots of constructive, critical and helpful comments. Most comments were carefully reflected in the revised manuscript. Our responses to individual comments are listed below.

**Major comments**

1. **It is not clear whether the only difference between SAM0 and CAM5 is the UNICON scheme. Is this the case? Even so, the scheme itself seems to contain many different changes, making it difficult to isolate cause and effect. There is an awful lot of speculation in the manuscript, from attributing increases in cloud fraction and liquid in the Arctic to increased condensation rate to attributing the increase in condensation rate to poleward transport of heat and moisture. Because it's not clear what exactly is different in SAM0 compared to CAM5 without reading the references in the manuscript in detail, I recommend that the authors conduct sensitivity tests to isolate the individual effects they are speculating. For example, if the authors claim that poleward moisture and heat transports are the main factors in SAM0 that cause an increase in condensation rate in the Arctic, then they could do sensitivity tests where they increase and decrease poleward moisture and heat transports in SAM0 by varying degrees to get a sense of whether or not they are dominant factors in affecting Arctic condensation rate.**

   → First of all, we are sorry for causing the confusion admitting that we didn't provide sufficient description on the models in the original manuscript. In this study, the only difference between SAM0 and CAM5 is the UNICON scheme, which is a unified convection scheme that replaces 1) CAM5's deep and shallow convection scheme (Park, 2014a, 2014b), and 2) convective detrainment process (Park et al., 2017). Other features such as dynamic core, cloud microphysics, PBL, etc. are exactly the same for the two models. We now carefully describe this in the revised manuscript.

   As the reviewer commented, the change of the convection scheme causes many different changes in the results between the two. To look at those differences more clearly, we show the evidence that SAM0 simulates the convection more strongly than CAM5, particularly in most of tropical Ocean and this is a direction to reduce bias from observation (Figure R1). Several previous studies have shown that enhanced convective activity in the Tropics enhances the poleward heat and moisture transport by inducing Rossby wave trains from Tropics toward pole that promote warm and moist advection from midlatitude into the Arctic (Lee et al. 2014; Fluorny et al. 2015).

   As with those studies, SAM0 captures Rossby wave trains emanating from Tropics better than CAM5 (Supplementary S1) and this leads to enhanced poleward heat and moisture transport in SAM0 (Figure 4 in the original manuscript). In SAM0, the increase in the poleward moisture

transport provides more water vapor source to form the cloud and the increase in the temperature by the enhanced poleward heat transport causes to product more cloud liquid condensation then cloud ice condensation in the Arctic region. All these results are consistent with the relative strengthening of convection in SAM0.

[Figure]

Figure R1. biases of annual-mean upward longwave (LW) radiative flux at TOA (FLUT) against the CERES–EBAF observation from (a) CAM5 and (b) SAM0 and (c) difference of FLUT between SAM0 and CAM5.

2. **The errors of the two observational datasets and Reanalysis data used are not discussed or addressed whatsoever. Please include a detailed description of the errors and biases in all three datasets. In particular, the GOCCP dataset does not account for lidar beam attenuation, which is particularly problematic in the Arctic, where optically thick supercooled liquid clouds attenuate the beam. Precipitating ice particles underneath these layers, which are known to commonly exist, would not be detected. If comparing the results of the models to GOCCP alone in terms of cloud amount, GOCCP might underestimate the actual cloud amount. I suggest that the authors either include a ground-based observational dataset to get an idea of the potential biases involved when comparing the models to GOCCP.**

→ Thanks to the reviewer's comment, we now provide descriptions about biases and errors of the satellite observations and the two reanalysis data used in this study especially over Arctic. As reviewer mentioned, CALIPSO-GOCCP may underestimate ice clouds in the lowest levels at midlatitudes and in polar regions because it's lidar beam may not detect some ice crystals underneath the optically thick stratocumulus clouds due to its attenuation (Cesana et al., 2015). CERES-EBAF also may produce an uncertainty over the Arctic, particularly for clear-sky retrievals due to the low albedo contrast between snow and clouds (English et al., 2014). We will carefully discuss the possible biases and errors in detail in Section 2.2 in the revised manuscript as you commented.

Regarding the use of GOCCP as a cloud amount estimate, we supplement the climatology data of long-term ground-based cloud and radiation measurements from 1998 to 2010 at the North Slope of Alaska (NSA) Barrow site (71.38N, 156.68W) from the Atmospheric Radiation

Measurement (ARM) Best Estimate (ARMBE) dataset (Xie et al. 2010) for the model evaluation (Figure R3).

3. **Although SAM0 is able to produce more low cloud amount and cloud liquid and less cloud ice as illustrated in Figure 2, it is not clear from the figures until Figures 6-9 how the models compare against observations. It could be that SAM0 overshoots low-cloud amount/cloud liquid or undershoots cloud ice relative to the observations. I would suggest including observations in Figure 2 as well. This could be done if the authors were to e.g. run the model in single column mode and compare their results with ground-based observations from the M-PACE field campaign. This could also provide additional evidence to support the authors' claims using an additional complementary ground-based observational dataset. This should also be clarified on lines 17-19 in the Abstract, where it should be specified what observational dataset the reduced biases are with respect to.**

→ Thanks to Reviewer #1's suggestion, we revised Figure 2 of the original manuscript by adding ERA-interim reanalysis cloud liquid/ice contents data (See Figure R2). We tried to use satellite observation data, but we could not find the data processing method. We inevitably used ERA-interim data. In the result, CAM5 underestimated the total cloud condensation with underestimating both cloud liquid and ice condensation against ERA-interim data. In the SAM0, although cloud ice condensation is underestimated as much as CAM5, cloud liquid condensation is simulated closed to observation, which reduces the overall bias of total cloud condensate in CAM5.

Instead of conducting a single-column model experiment using M-PACE field campaign data, the climatology data of long-term ground-based cloud and radiation measurements from 1998 to 2010 at the North Slope of Alaska (NSA) Barrow site (71.38N, 156.68W) from the Atmospheric Radiation Measurement (ARM) Best Estimate (ARMBE) dataset (Xie et al. 2010) are used for the model evaluation (Figure R3). In the result, the total cloud fraction (TAC) in CAM5 was simulated totally less than in the observation except for July and August, and the liquid water path (LWP) also underestimate over the entire period. Accordingly, the downward shortwave flux was overestimated, and the downward longwave flux was underestimated particularly in autumn and winter. Although TAC in SAM0 was slightly overestimated in the summertime compared to the observation, SAM0 reduced the bias of CAM5 in the rest of the period. The LWP and the surface radiation fluxes were also simulated closer to the observation than in CAM5.

[Figure]

**Figure R2. Annual-mean vertical profiles of grid-mean (a) cloud condensate mass (cloud liquid + cloud ice), (b) cloud liquid mass, and (c) cloud ice mass averaged over the Arctic area from ERA-interim (ERA, black line), SAM0 (red lines), and CAM5 (blue lines) and (d) the difference of cloud fraction between SAM0 and CAM5**

[Figure]

**Figure R3. Annual cycles of total cloud fraction (TAC, upper in (a)), liquid water path (LWP, bottom in (a)), surface downward shortwave flux (FSDS, upper in (b)), and surface downward longwave flux (FLDS, bottom in (b)) from the climatology of ground-based cloud and radiation measurements at North Slope of Alaska (NSA) Barrow site, (black line), SAM0 (red line), and CAM5 (blue line).**

4. **Figure 3: Why these microphysical tendencies? Why not include other microphysical processes such as accretion, autoconversion, wet/dry deposition as well? This analysis may be missing**

processes that are more important than net condensation rate. Also, the nonlinear interactions between the model tendencies are not quantified in Figure 3; the various processes all feedback and are dependent on one another. The authors could make this analysis more rigorous by quantifying the contribution of these liquid and ice tendencies to cloud liquid and ice mass using a multiple linear regression approach.

→ Those microphysical tendencies such as accretion are already included in the PRS (cyan lines) in figure 3 of the original manuscript and they obviously sink the cloud condensation for both liquid and ice processes. Aerosol wet/dry deposition could not be included in the budget analysis because the process is not a cloud liquid/ice condensation process. We agree that these microphysical processes simultaneously feedback from each other in the real world. In the model used in this study, however, each process is conducted sequentially (Park et al. 2014), so that each cloud microphysical tendency can be calculated separately, and the sum of them is close to zero.

5. **The strong negative bias in TCA seems to persist in the summer (Figure 8), yet why does there appear to be little to no SWCF bias? Is LCA more relevant than TCA?**
→ Does reviewer mean Greenland region? If so, the Greenland region in the model is an area of over 1 m of snow, and albedo is close to 1. Therefore, the area reflects most of the incoming shortwave regardless of cloud amount, and SWCF is close to zero.

**Minor comments**

1. **Page 9: LCA was never explicitly defined. I'm assuming this stands for low-cloud amount. Does this include clouds from 700hPa to 1000 hPa?**
→ We are sorry for causing confusion. LCA stands for low-cloud fraction, which, as reviewer commented, means a cloud fraction between 700 hPa and 1000 hPa. The definition of LCA was added to the revised manuscript.

2. **Section 2.1: What is the vertical resolution of the model? What was used as the spin-up time of the model? Which COSP simulator was used (e.g. was it the lidar simulator?)**
→ Both SAM0 and CAM5 have 30 vertical layers. For model experiments, we follow the standard procedure of AMIP-type experiment, which does not require spin-up time. Yes, we used the lidar simulator in the COSP We added this point in the revised manuscript.

3. **Figure 3: Does the WBF process include snow?**
→ In both models, the WBF process only implies conversion from liquid to ice.

4. **Section 3.2: The sentences referring to the temperature inversions are written in a confusing way. It seems like the temperature inversions should be mentioned after the effect of LCA on surface fluxes, not before since it's a consequence of the clouds.**

→ We are sorry for the confusion. We changed the sentence as: "In the Arctic during winter, less LCA in CAM5 reduces FLUT over the land and the sea-ice region in the lower troposphere in a circumstance which the temperature in the cloudy layer is higher than that at the surface (i.e., temperature inversion). The less LCA also reduces downward LW radiation at the surface (FLDS), which leads to colder near-surface air than the observation, resulting in enhancement of the temperature inversion."

5. **Rather than state that "most of the shaded area" is statistically significant, why not shade the statistically insignificant areas to avoid crowding the plot?**

→ In the revised manuscript, only the statistically significant areas were plotted.

Response to Reviewer #2

We sincerely appreciate Reviewer #2 for spending his/her invaluable time to give us lots of constructive, critical and helpful comments. Most comments were carefully reflected in the revised manuscript. Our responses to individual comments are listed below.

**Major comments**

1. **For the link between statement 3 and 4, the authors cite Park et al. (2014): "the horizontal and vertical transports of heat and moisture are the important factors inducing the net condensation of water vapor into cloud liquid (NCD) both in SAM0 and CAM5". Are these factors also important in the Arctic regions? How dominant are these factors? Are there any other important factors? Could other modifications in SAM0 contribute to the enhancement of NCD? For example, in-cloud turbulence and precipitation from super-cooled liquid clouds.**
   **The following analysis of concurrence and correlations between advection and clouds are not definite evidence either. As a model study (instead of observational data analysis), more concrete evidence is expected for this imperative link. How about adding a budget analysis like the one for statement 2 and 3 or experiments turning on/off certain model processes?**
   → As reviewer mentioned, we examined all of the cloud liquid and ice condensate process tendencies including in-cloud turbulences and precipitation in each grid-box of the model in the Arctic region (in figure 3 in the original manuscript). By this effort, we found that net condensation of water vapor into cloud liquid (NCD) is a single dominant process for the Arctic cloud liquid condensation compared to other physical processes (e.g., nonlocal asymmetric turbulent eddies (CON), local symmetric turbulent eddies (PBL), precipitation (PRS), conversion of cloud liquid into cloud ice (WBF), etc.). Given the cloud fraction, in the model, the NCD is explicitly calculated by the saturation equilibrium in cloud macrophysics scheme, which indicates that the NCD in a grid-box is produced more with the condition of more water vapor and lower grid-mean temperature. The water vapor source can be increased by the convergence of moisture transport and the grid-mean temperature can be changed by various physical and dynamical processes such as radiation, grid-scale advection, moist turbulence, etc.. In this study, although SAM0 has a higher temperature than CAM5 in the Arctic, the net condensate rate in SAM0 is larger than in CAM5 because of relatively larger poleward moisture transport into the Arctic. This point indicates that moisture transport is a dominant factor for the generation of NCD. We explained this point clearly in the revised manuscript. Although we agree with the Reviewer #2, however, there is a difficulty to determine which factor indeed contributes to the increase in the NCD through the budget analysis, because NCD is calculated by a complex relationship between temperature, water vapor, cloud fraction in a

grid-box (i.e., saturation equilibrium). Instead of additional budget analysis, we provide more evidence that SAM0 exhibits stronger convection (Figure. R1) and explanations on how this can be related to the enhanced poleward moisture transport.

2. **The authors speculate that horizontal advection rather than vertical transport is responsible for the enhancement of NCD due to the identical surface boundaries. Could the authors provide some discussion instead of speculation on the role of vertical transport? Is it possible that the modification of SAM0 on the convection scheme alters the vertical structure of the atmosphere, which could also contribute to the NCD enhancement?**

→ As the reviewer #2 commented, the replacement of convection scheme (i.e., UNICON scheme) in SAM0 can change the vertical structure of atmosphere as shown in Figure 4d a, h in the original manuscript. As a response to the comment, we examine the relative amount of the meridional moisture flux and vertical moisture flux averaged at 65° N (Figure R4). Figure R4 shows each flux difference between SAM0 and CAM5. For a fair comparison, we converted the vertical moisture flux unit to [g kg$^{-1}$ m$^{-1}$ s$^{-1}$]. The results indicate that the horizontal moisture flux is about 1000 times larger than vertical moisture flux. We address this in the revised manuscript.

[Figure]

**Figure R4. Differences the zonal-mean (a) meridional moisture flux and (b) vertical moisture flux at 65° N between SAM0 and CAM5**

3. **Evaluated against ERA-Interim, SAM0 exhibits smaller biases than CAM5 in annual-mean geopotential heights simulations. However, this does not guarantee a more accurate simulation of heat and moisture advection. For example, SAM0 overestimates geopotential height at 850 hPa in tropical Pacific and underestimates in sub-Arctic Pacific, leading to larger heat transport into the Arctic. The moisture transport is also affected by the location of pathways. Could the authors evaluate the heat and moisture advection directly against observations? More importantly, this is the only way to validate the claim of "proper heat and moisture transport is the key process in**

**simulating Arctic climate".** Otherwise, the evidence could only support a claim of "enhanced heat and moisture transport improve Arctic climate simulations" because the larger advection could be an overcompensation in order to increase liquid clouds.

→ We agree with reviewer's comments. From the additional calculation, we found that the SAM0 reduces the bias of the poleward heat and moisture transports against observation compared to the CAM5. We calculated the biases of poleward heat and moisture transports in both CAM5 and SAM0 against ERA-Interim reanalysis (Figure R5 and R6). CAM5 overestimates both moisture and heat fluxes over the midlatitude region against the observation but underestimates those on the periphery (around 70° N) of the Arctic circle. Although the positive bias over the midlatitude region still remains, SAM0 reduces the biases of CAM5 on the periphery (around 70° N) of the Arctic circle. The horizontal pattern also shows that the enhanced poleward moisture transport in SAM0 is in agreement with the observation. Figure R7 shows the vertically integrated annual mean moisture flux and its convergence from ERA-Interim and from those differences between SAM0 and CAM5. The poleward moisture flux in SAM0 is entirely enhanced in the midlatitude and subpolar regions compared with CAM5 as shown in the original manuscript Figure 4. The poleward moisture flux increases particularly in the North Atlantic and North American regions where are similar to the regions with large poleward moisture transport in ERA-Interim. We explained this point clearly in the revised manuscript.

[Figure]

**Figure R5. CAM5 biases of zonal-mean meridional fluxes of (a, b, and c) moisture and (d, e, and f) heat by (a and d) total processes (i.e., the transported sum by mean meridional circulation,**

stationary eddies, and transient eddies), (b and e) mean meridional circulation (MMC), and (c and f) transient eddies (TE) against the ERA-interim reanalysis. The black lines in (a) and (d) denote the bias of zonal-mean convergence of total moisture flux in $10^{-7}$ g kg$^{-1}$ s$^{-1}$ and total heat flux in $10^{-5}$ K s$^{-1}$, the black lines in (b) and (e) denote the bias of zonal mean meridional wind in m s$^{-1}$, and the black lines in (c) and (f) denote the bias of zonal-mean zonal wind in m s$^{-1}$ against the ERA-interim reanalysis, respectively.

[Figure]

**Figure R6. Identical with Figure R5 except for SAM0.**

[Figure]

**Figure R7. Vertically-integrated annual-mean moisture flux in m g kg$^{-1}$ s$^{-1}$ (arrow) and its convergence (shaded) from (a) ERA-Interim and from (b) those differences between SAM0 and CAM5. Black contour denotes the Arctic circle (65° N).**

4. **Page 1 Line 23-26, "proper simulation of poleward heat and moisture transport is one key factor for simulating Arctic clouds" could not be wrong. However, drawing this conclusion from an uncertainty level of "association" hurts rather than lends credibility, especially in a model study. Would the authors provide more evidence to substantiate this claim? Or this manuscript could focus on the improvement of SAM0 and thoroughly evaluate all the causes.**

→ We agree with the Reviewer. Although in this revision, we provide more pieces of evidence that poleward transports are tightly related with the increase of NCD and proper simulation of the poleward transports reduce the bias, our results are only from the comparison between specific two models. Obviously, the results cannot be generalized to other models. Accordingly, we tried to tone down the sentence and other concluding remarks in the revised manuscripts.

**Minor comments**

14. **Page 6 Figure 3, have the authors looked at the seasonal breakdown? In winter, could larger moisture transport lead to more NCD to cloud ice?**

→ We examined the seasonal cycles of poleward moisture transport and NCD for both cloud liquid and ice condensate rate (Figure R8). In both the two models, the NCD for cloud liquid (NCD-liq) seasonal variabilities are nearly identical to those of poleward moisture transport, which also indicates a positive relationship between those. Throughout the year, SAM0 simulates more poleward moisture transport into the Arctic region than CAM5, which increases NCD-liq not the NCD for cloud ice (NCD-ice). The difference of NCD-liq is in line the difference of relative humidity (RH), which denotes that the NCD-liq are calculated by saturation equilibrium process, as mentioned above. In wintertime, the NCD-ice in CAM5 is larger than NCD-liq, but SAM0 simulates more NCD-liq than NCD-ice throughout the year. We added this figure and description in the revised manuscript.

[Figure]

**Figure R8. Annual cycles of zonal-mean poleward moisture transport (PMT65) at 65° N, net condensation rate of water vapor into cloud liquid (NCD-liq, center solid line), and net condensation rate of water vapor into cloud ice (NCD-ice, center dashed line) averaged over the Arctic area from SAM0 (red line) and CAM5 (blue line) and the differences of NCD-liq (black bars) and relative humidity (green bars)**

15. **Page 8 Figure 4, what is the percentage of these differences relative to the absolute values of CAM5. If it's too messy to plot on the figure, a description in the text would be fine too.**

→ The difference of poleward moisture (heat) transport between SAM0 and CAM5 is about 10% (15%) of its climatology, respectively. We added this information in the revised manuscript.

17. **Page 9 Figure 5 a and b, the correlation seems to weaken in the recent years. Any reasons?**

→ We couldn't find out the reasons. We think that the issue is an interesting subject in the future work.

20. **Page 10 Figure 6c, I am confused about the unit: $10^3$ K kg m$^{-1}$ s$^{-1}$. Could it be $10^3$ Km$^{-1}$ s$^{-1}$ ?**

→ Yes. $10^3$ K m$^{-1}$ s$^{-1}$ is correct. We corrected the unit in the revised manuscript.

21. **Most of the paragraphs lack a topic sentence to guide readers. For example, the topic sentence for the first paragraph in Section 3 could be "SAM0 reduces the negative biases of CAM5 in liquid cloud simulations".**

→ Thanks. We entirely corrected paragraphs in the revised manuscript.

**Other minor issues**

→ We sincerely thank for reviewer's kind comments. All of the minor issues are fully considered in the revised manuscripts. In addition, some other errors are corrected.

Response for Reviewer #3

We sincerely appreciate Reviewer 3 for spending his/her invaluable time to give us lots of constructive, critical and helpful comments. Most comments were carefully reflected in the revised manuscript. Our responses to individual comments are listed below.

**Major comments**

1. **Improvements of simulated Arctic clouds: according to Figure 1, although SAM0 cloud fraction is closer to observations, significant biases still persist, especially in winter. The improvement in PR90 is marginal. I suggest the authors to also compare the liquid and ice water path to the observations (e.g. Lenaerts et al. 2017), because they are also important for cloud radiative effects. For example, does the decrease in cloud ice mass (Figure 2c) make it closer to observations?**
   → As reviewer's suggestion, we added liquid water path (LWP) and ice water path (IWP) bias against from ERA-interim data (Figure R9). Additionally, we added ERA-interim data cloud contents in the original manuscript Figure 2 (Figure R2). We tried to use satellite observation data as shown in Lenaerts et al. (2017), but we could not find the data processing method, so I inevitably used ERA-interim data. Lenaerts et al. (2017) showed the LWP and IWP from CloudSat-Calipso climatology (C-C in their paper) and the difference between ERA-Interim and C-C. We could speculate the comparison between the model results and C-C, indirectly. In our result, CAM5 considerably underestimates the total cloud content with underestimating both cloud liquid and ice amount against ERA-interim data. In the SAM0, Although cloud ice condensate is underestimated as much as CAM5, cloud liquid condensate is simulated closed to observation, which reduces the overall bias of CAM5 in total cloud condensate. We mentioned the point in detail in the revised manuscript.

[Figure]

**Figure R9. Annual mean liquid water path (LWP) (upper panel) and ice water path (IWP) (bottom panel) of (first column) ERA interim data and the differences of (second column) CAM5 and (third column) SAM0 from ERA interim data**

2. **Relationship between meridional fluxes and increased cloud liquid:**

   **a) Is vertical advection included in the meridional transport? Heat and moisture transport into the Arctic doesn't just happen in the horizontal plane. In fact, eddies transport moisture along (moist) isentropes.**

   → We agree with the reviewer. To provide the best answer for your question, we examine the relative amount of the meridional moisture flux and vertical moisture flux averaged at 65° N (Figure R4). Figure R4 shows each flux difference between SAM0 and CAM5. For a fair comparison, we convert the vertical moisture flux unit to [g kg$^{-1}$ m$^{-1}$ s$^{-1}$]. The results indicate that the meridional moisture flux is about 1000 times larger than vertical moisture flux. So, we think the vertical moisture flux can be ignored in this study. We explained this point more clearly in the revised manuscript.

   **b) Even though increased moisture flux and Arctic liquid cloud are correlated, it doesn't provide causation.**

→ We agree with the reviewer's comments. We think we do not provide sufficient explanation on the link between poleward moisture transport and the Arctic net condensation of water vapor into cloud liquid (NCD) in the original manuscript. In the model, given the cloud fraction, the NCD is explicitly calculated by the saturation equilibrium in cloud macrophysics scheme, which indicates that the NCD in a grid-box is produced more with the condition of more water vapor and lower grid-mean temperature. The water vapor source can be increased by the convergence of moisture transport and the grid-mean temperature can be changed by various physical and dynamical processes such as radiation, grid-scale advection, moist turbulence, etc.. In this study, although SAM0 has a higher temperature than CAM5 in the Arctic, the net condensate rate in SAM0 is larger than in CAM5 because of relatively larger poleward moisture transport into the Arctic. This point indicates that moisture transport is a dominant factor for the generation of NCD. We explained this point clearly in the revised manuscript.

We agree that the correlation between poleward moisture transport and Arctic NCD do not explain definite causality. In the revised manuscript, we discuss this issue shortly for proper interpretation and focus on to explain more clearly the link between poleward moisture transport and NCD with analyzing seasonal variabilities between them and horizontal pathway of poleward moisture transport. Additionally, we provide more evidence that SAM0 exhibits stronger convection (Figure. R1) and explanations on how this can be related to the enhanced poleward moisture transport.

3. **The results shown here are from atmospheric only GCMs. Since the results depend on the atmospheric heat transport, ocean coupling could potentially alter the results. Have the authors looked at whether the changes in heat and moisture fluxes are still robust in coupled SMA0?**

→ Thank you for your constructive suggestion. We agree to need the additional experiment using a fully-coupled model owing to air-sea interaction. The factor may be more important in recent years when the Arctic has undergone rapid warming and Arctic sea ice has drastically declined. Currently, we do not have the present-day full-coupled experiment data set of both CAM5 and SAM0. We will try this subject in future work.

**Minor comments**

1. **Please hatch the maps to show their significance level, instead of saying "most shaded areas exceed a 95% significance level". Since not all areas are significant, it is useful to know where it is not significant.**

→ We agree with reviewer's comment. In the revised manuscript, only the statistically significant areas were plotted.

2. **P3 L7: What microphysics scheme does SAM0 use? Is it the same for CAM5? If not, it can introduce additional sensitivity.**

→ Both CAM5 and SAM0 use the same microphysics scheme. In this study, the only difference between SAM0 and CAM5 is the UNICON scheme. The UNICON scheme is a unified convection scheme that replaces the CAM5's deep and shallow convection scheme (Park, 2014a, 2014b), and the convective detrainment process (Park et al., 2017).

3. **P3 L23: Are different periods from 1979 to 2015 selected to compare with the corresponding observations (CALIPSO and CERES)? If so, it should be specified, since the simulation period is much longer than what the observations cover. If not, are the results sensitive to the mismatch in periods?**

→ Yes. The integration period of both CAM5 and SAM0 is 36 years from January 1979 to February 2015 and we use the climatology of the CALIPSO-GOCCP from June 2006 to November 2010 and the climatology of CERES-EBAF from March 2000 to February 2013. The period chosen is limited by the availability of each satellite observation data. Although the period among the model outputs and the satellite observations are different, we think that the data could be comparable when using each climatology. To verification of this point, we plotted same figures (Figure 1a, Figure 7d,e,f, and Figure 8d,e,f in the original manuscript) using model data averaged with the same period to each observation data (Figure R10 and R11). The results show almost the same as the figures averaged with whole period.

[Figure]

Figure R10. Identical with Figure 1 in the original manuscript except for adding the results averaged from June 2006 to November 2010 (dashed line) for each model

[Figure]

Figure 11. (a), (b), and (c) are identical with Figure 7d,e,f in the original manuscript and (d), (e), and (f) are identical with Figure 8d,e,f in the original manuscript except for using data averaged from March 2000 to February 2013 for each model

4. **P4 L14-15: The TCA bias also varies seasonally. SAM0's improvement is the most significant in summer, but less so in winter. Have the authors investigated the seasonal and spatial variability in poleward moisture transport? The seasonal cycle combined with spatial maps could shed light on why clouds are underestimated in the Arctic.**

→ As reviewer's comment, we investigated the seasonal and spatial variability of poleward moisture transport and NCD in the response manuscript (Figure R7 and R8). In an aspect of seasonal variability, the poleward moisture transport in SAM0 is simulated more than CAM5 throughout the year (Figure R8). In both the two models, the poleward moisture fluxes are the largest from summer to autumn, given that eddy activity is the largest in summer and mean meridional circulation is the largest in winter in the high latitude in the Northern Hemisphere. The associated NCD averaged over the Arctic region is nearly identical to the poleward moisture transport. The seasonal variability of the difference of NCD averaged over the Arctic region is almost coincident with that of the grid-mean RH, which explains why the Arctic liquid

cloud amount increases from May to September as shown in Figure 1 in the original manuscript. Figure R7 shows the vertically-integrated annual-mean moisture flux and its convergence from ERA-Interim and from those differences between SAM0 and CAM5. the poleward moisture flux in SAM0 is entirely enhanced in the midlatitude and subpolar regions compared to CAM5 as shown in the original manuscript Figure 4. The poleward moisture flux increases particularly in the North Atlantic and North American regions where are identical with the regions with large poleward moisture transport in ERA-Interim. We added this point in the revised manuscript.

5. **P6 Figure 3: Are these budgets closed? Net tendency profiles for liquid and ice from each model can be added to these figures.**

    → Actually, there was a simple mistake in calculating the NCD for the cloud liquid in the original manuscript Figure 3a. Although a little bit more value of NCD for the cloud liquid is calculated, it did not affect the overall description about Figure 3. After we fixed it in the revised manuscript, the sum of all tendencies for both cloud liquid and ice is closed to zero. So we think that it is not necessary to add the zero line in Figure 3. But we described this point in text in the revised manuscript.

6. **P8 Figure 4: According to the moisture and heat flux convergence, we expect increased liquid condensation thus more liquid clouds at around 70N. Is this the case? For example, in winter the total cloud increase is quite spatially uniform over the Arctic Ocean (Fig 7b).**

    → The TCA increase in SAM0 during wintertime is relatively larger on the periphery of the Arctic circle (between 60° N and 70° N) than the Arctic inside region. The TAC increases particularly in the North Atlantic and North American regions where are identical with the regions with the greatest increase in the poleward moisture transport as shown in Figure R7b in the response manuscript.

7. **P8 L22-P9 L2: We all know that correlation does not mean causation. By just showing correlation, the causality is not proven. For example, it is also possible that both changes are caused by a third factor that is not in the analysis.**

    → We agree with your comment. We described this point previously in major comment 2-(b).

8. **P8 L13: LCA was never spelled out in the paper.**

    → We are sorry for the confusion. LCA stands for the low-cloud fraction. The definition of LCA has been added to the revised manuscript.

9. **P9 L14: What makes these two models the outliers? Are there any physical reasoning to say so? One should not just pick and choose the models or discard the end members because they do not agree with your hypothesis.**

→ We would like to use all available CMIP5 models including the necessary variables. Although bcc-csm1-1-m and MPI-ESM-LR models are outliers in the relationship between the meridional moisture transport and TCA and LCA, the models are in-line in the relationship between the meridional heat transport and LWP ration.

10. **P10 Figure 6: Are the widths of black lines in (a) and (b) represent the spread of observed poleward moisture transport? If not, these lines are very misleading and unnecessary.**

→ The widths of black lines in the original manuscript Figure 6 means not "the spread of observed poleward moisture transport" but "just values of CALIPSO cloud fraction". We removed the line and pointed out the "values of CALIPSO cloud fraction" at y-axis in the revised manuscript.

11. **P10 L7: Upward LW at TOA includes both clear sky and cloud effects. Have the authors look at the cloud radiative effect differences between the two models? The negative bias in upward LW at TOA in the Atlantic sector seems to get worse in SAM0. Is it because in this region, SAM0 is producing too much clouds comparing to the observations?**

→ We checked the longwave cloud radiative forcing (LWCF) during the wintertime and found that the LWCF is a positive bias in the Atlantic sector while negative bias in the other land and sea-ice region. There is temperature inversion over the land and sea-ice region, in contrast, it does not exist over the Arctic open-sea region (i.e., Atlantic sector) because the sea surface temperature is warm. Thus increase in the cloud amount over the open-sea region reduce the FLUT in SAM0, which enhanced the negative bias of FLUT in CAM5.

12. **P10 L11: Figure 7 shows TCA, but the argument the authors give here involves LCA. Is the LCA change the same as TCA?**

→ We plotted the TCA in Figure 7 in the original manuscript for consistency in the entire manuscript. The LCA change is almost the same as TCA. We described this point in text in the revised manuscript.

13. **P11 L7: Remove "a" before "summertime biases"**

→ Thanks. We corrected in the revised manuscript.

14. **P13 Figure 8: Does panel d) suggest that CAM5 SW cloud forcing is too weak at the surface? This would lead to a warm bias over the Arctic ocean. But g) shows a cold bias, which means that the**

**LW cloud forcing bias (not enough warming at surface) dominates the net forcing. It would be helpful to see both LWCF and SWCF to get a fuller picture. I suggest the authors to plot LWCF and SWCF for both TOA and surface, at least to include them in the supplementary material.**

→ We totally agree with reviewer's comment. During summertime, the surface temperature response is not explained by only SWCF. Rather the surface temperature is determined by surface net flux of SW and LW. So we plotted the net SW radiation at the surface (FSNS) and net LW radiation at the surface (FLNS) and the sum of FSNS and FLNS in the original manuscript Figure 9. The difference of the sum of FSNS and FLNS between SAM0 and CAM5 agrees with the difference of near-surface temperature ($T_{2m}$). In addition, we add the plots of SWCF and LWCF at TOA (there is not SWCF and LWCF at the surface) in the supplementary material (Figure R12).

[Figure]

**Figure R12. Biases of (upper) TCA against the CALIPSO–GOCCP observation, (middle) shortwave cloud forcing at TOA (SWCF) and (lower) longwave cloud forcing at TOA (LWCF)**

**against the CERES–EBAF observation during JJA obtained from (left) CAM5 and (right) SAM0; and (center) differences of each variable between SAM0 and CAM5.**

15. **P14 L11: The wording "SAM0 remedies these problems" is too strong, given that significant cloud biases still persist.**

→ We agree with reviewer. We corrected inappropriate sentences not to describe too strongly in the revised manuscript.

**Reference**

Cesana, G., Waliser, D. E., Jiang, X. and Li, J.-L. F.: Multi-model evaluation of cloud phase transition using satellite and reanalysis data, J. Geophys. Res. Atmos., (JUNE), n/a-n/a, doi:10.1002/2014JD022932, 2015.

Flournoy, M. D., Feldstein, S. B., Lee, S., & Clothiaux, E. E. (2016). Exploring the Tropically Excited Arctic Warming Mechanism with Station Data: Links between Tropical Convection and Arctic Downward Infrared Radiation. Journal of the Atmospheric Sciences, 73(3), 1143–1158. https://doi.org/10.1175/JAS-D-14-0271.1

English, J. M., Kay, J. E., Gettelman, A., Liu, X., Wang, Y., Zhang, Y. and Chepfer, H.: Contributions of clouds, surface albedos, and mixed-phase ice nucleation schemes to Arctic radiation biases in CAM5, J. Clim., 27(13), 5174–5197, doi:10.1175/JCLI-D-13-00608.1, 2014.

Lee, S., & Yoo, C. (2014). On the causal relationship between poleward heat flux and the equator-to-pole temperature gradient: A cautionary tale. Journal of Climate, 27(17), 6519–6525. https://doi.org/10.1175/JCLI-D-14-00236.1

Park, S., Bretherton, C. S. and Rasch, P. J.: Integrating Cloud Processes in the Community Atmosphere Model, Version 5, J. Clim., 27(18), 6821–6856, doi:10.1175/JCLI-D-14-00087.1, 2014

Xie, and Coauthors, 2010: Clouds and more: ARM climate modeling best estimate data. Bull. Amer. Meteor. Soc., 91, 13–20

---

## Author Response (AR1)

Response to Reviewer #1

We sincerely appreciate Reviewer #1 for spending his/her invaluable time to give us lots of constructive, critical and helpful comments. Most comments were carefully reflected in the revised manuscript. Our responses to individual comments are listed below.

**Major comments**

1. **It is not clear whether the only difference between SAM0 and CAM5 is the UNICON scheme. Is this the case? Even so, the scheme itself seems to contain many different changes, making it difficult to isolate cause and effect. There is an awful lot of speculation in the manuscript, from attributing increases in cloud fraction and liquid in the Arctic to increased condensation rate to attributing the increase in condensation rate to poleward transport of heat and moisture. Because it's not clear what exactly is different in SAM0 compared to CAM5 without reading the references in the manuscript in detail, I recommend that the authors conduct sensitivity tests to isolate the individual effects they are speculating. For example, if the authors claim that poleward moisture and heat transports are the main factors in SAM0 that cause an increase in condensation rate in the Arctic, then they could do sensitivity tests where they increase and decrease poleward moisture and heat transports in SAM0 by varying degrees to get a sense of whether or not they are dominant factors in affecting Arctic condensation rate.**

→ First of all, we are sorry for causing the confusion admitting that we didn't provide sufficient description on the models in the original manuscript. In this study, the only difference between SAM0 and CAM5 is the UNICON scheme, which is a unified convection scheme that replaces 1) CAM5's deep and shallow convection scheme (Park, 2014a, 2014b), and 2) convective detrainment process (Park et al., 2017). Other features such as dynamic core, cloud microphysics, PBL, etc. are exactly the same for the two models. We described this on lines 80-81 in the revised manuscript.

As the reviewer commented, the change of the convection scheme causes many different changes in the results between the two. To look at those differences more clearly, we show the evidence that SAM0 simulates the convection more strongly than CAM5, particularly in most of tropical Ocean and this is a direction to reduce bias from observation (Supplementary S7 in the revised manuscript). Several previous studies have shown that enhanced convective activity in the Tropics enhances the poleward heat and moisture transport by inducing Rossby wave trains from Tropics toward pole that promote warm and moist advection from midlatitude into the Arctic (Lee et al. 2014; Fluorny et al. 2015).

As with those studies, SAM0 captures Rossby wave trains emanating from Tropics better than CAM5 (Supplementary S6c) and this leads to enhanced poleward heat and moisture transport in SAM0 (Figure 5 in the revised manuscript). In SAM0, the increase in the poleward moisture

transport provides more water vapor source to form the cloud and the increase in the temperature by the enhanced poleward heat transport causes to product more cloud liquid condensation then cloud ice condensation in the Arctic region. All these results are consistent with the relative strengthening of convection in SAM0 (lines 230-236 in the revised manuscript).

2. **The errors of the two observational datasets and Reanalysis data used are not discussed or addressed whatsoever. Please include a detailed description of the errors and biases in all three datasets. In particular, the GOCCP dataset does not account for lidar beam attenuation, which is particularly problematic in the Arctic, where optically thick supercooled liquid clouds attenuate the beam. Precipitating ice particles underneath these layers, which are known to commonly exist, would not be detected. If comparing the results of the models to GOCCP alone in terms of cloud amount, GOCCP might underestimate the actual cloud amount. I suggest that the authors either include a ground-based observational dataset to get an idea of the potential biases involved when comparing the models to GOCCP.**

→ Thanks to the reviewer's comment, we now provide descriptions about biases and errors of the satellite observations and the two reanalysis data used in this study especially over Arctic. As reviewer mentioned, CALIPSO-GOCCP may underestimate ice clouds in the lowest levels at midlatitudes and in polar regions because it's lidar beam may not detect some ice crystals underneath the optically thick stratocumulus clouds due to its attenuation (Cesana et al., 2015). CERES-EBAF also may produce an uncertainty over the Arctic, particularly for clear-sky retrievals due to the low albedo contrast between snow and clouds (English et al., 2014). We will carefully discuss the possible biases and errors in detail on lines 107-119 in the revised manuscript.

Regarding the use of GOCCP as a cloud amount estimate, we supplement the climatology data of long-term ground-based cloud and radiation measurements from 1998 to 2010 at the North Slope of Alaska (NSA) Barrow site (71.38N, 156.68W) from the Atmospheric Radiation Measurement (ARM) Best Estimate (ARMBE) dataset (Xie et al. 2010) for the model evaluation (Figure 2 in the revised manuscript). (lines 119-125 at the revised manuscript)

3. **Although SAM0 is able to produce more low cloud amount and cloud liquid and less cloud ice as illustrated in Figure 2, it is not clear from the figures until Figures 6-9 how the models compare against observations. It could be that SAM0 overshoots low-cloud amount/cloud liquid or undershoots cloud ice relative to the observations. I would suggest including observations in Figure 2 as well. This could be done if the authors were to e.g. run the model in single column mode and compare their results with ground-based observations from the M-PACE field campaign. This could also provide additional evidence to support the authors' claims using an**

**additional complementary ground-based observational dataset. This should also be clarified on lines 17-19 in the Abstract, where it should be specified what observational dataset the reduced biases are with respect to.**

→ Thanks to Reviewer #1's suggestion, we revised Figure 2 of the original manuscript by adding ERA-interim reanalysis cloud liquid/ice contents data (Supplementary Figure S2 in the revised manuscript). We tried to use satellite observation data, but we could not find the data processing method. We inevitably used ERA-interim data. In the result, CAM5 underestimated the total cloud condensation with underestimating both cloud liquid and ice condensation against ERA-interim data. In the SAM0, although cloud ice condensation is underestimated as much as CAM5, cloud liquid condensation is simulated closed to observation, which reduces the overall bias of total cloud condensate in CAM5. (lines 161-165 in the revised manuscript) Instead of conducting a single-column model experiment using M-PACE field campaign data, the climatology data of long-term ground-based cloud and radiation measurements from 1998 to 2010 at the North Slope of Alaska (NSA) Barrow site (71.38N, 156.68W) from the Atmospheric Radiation Measurement (ARM) Best Estimate (ARMBE) dataset (Xie et al. 2010) are used for the model evaluation (Figure 2 in the revised manuscript). In the result, the total cloud fraction (TAC) in CAM5 was simulated totally less than in the observation except for July and August, and the liquid water path (LWP) also underestimate over the entire period. Accordingly, the downward shortwave flux was overestimated, and the downward longwave flux was underestimated particularly in autumn and winter. Although TAC in SAM0 was slightly overestimated in the summertime compared to the observation, SAM0 reduced the bias of CAM5 in the rest of the period. The LWP and the surface radiation fluxes were also simulated closer to the observation than in CAM5. (lines 147-155 in the revised manuscript)

4. **Figure 3: Why these microphysical tendencies? Why not include other microphysical processes such as accretion, autoconversion, wet/dry deposition as well? This analysis may be missing processes that are more important than net condensation rate. Also, the nonlinear interactions between the model tendencies are not quantified in Figure 3; the various processes all feedback and are dependent on one another. The authors could make this analysis more rigorous by quantifying the contribution of these liquid and ice tendencies to cloud liquid and ice mass using a multiple linear regression approach.**

→ Those microphysical tendencies such as accretion are already included in the PRS (cyan lines) in figure 3 of the original manuscript (Figure 4 in the revised manuscript) and they obviously sink the cloud condensation for both liquid and ice processes. Aerosol wet/dry deposition could not be included in the budget analysis because the process is not a cloud liquid/ice condensation process.

5. **The strong negative bias in TCA seems to persist in the summer (Figure 8), yet why does there appear to be little to no SWCF bias? Is LCA more relevant than TCA?**

→ Does reviewer mean Greenland region? If so, the Greenland region in the model is an area of over 1 m of snow, and albedo is close to 1. Therefore, the area reflects most of the incoming shortwave regardless of cloud amount, and SWCF is close to zero.

**Minor comments**

1. **Page 9: LCA was never explicitly defined. I'm assuming this stands for low-cloud amount. Does this include clouds from 700hPa to 1000 hPa?**

→ We are sorry for causing confusion. LCA stands for low-cloud fraction, which, as reviewer commented, means a cloud fraction between 700 hPa and 1000 hPa. The definition of LCA was added on line 263 in the revised manuscript.

2. **Section 2.1: What is the vertical resolution of the model? What was used as the spin-up time of the model? Which COSP simulator was used (e.g. was it the lidar simulator?)**

→ Both SAM0 and CAM5 have 30 vertical layers. For model experiments, we follow the standard procedure of AMIP-type experiment, which does not require spin-up time. Yes, we used the lidar simulator in the COSP We added these points on line 98 and 101 in the revised manuscript.

3. **Figure 3: Does the WBF process include snow?**

→ In both models, the WBF process only implies conversion from liquid to ice.

4. **Section 3.2: The sentences referring to the temperature inversions are written in a confusing way. It seems like the temperature inversions should be mentioned after the effect of LCA on surface fluxes, not before since it's a consequence of the clouds.**

→ We are sorry for the confusion. We changed the sentence as: "In the Arctic during winter, less LCA in CAM5 reduces FLUT over the land and the sea–ice region in the lower troposphere because the temperature in the cloudy layer is higher than that at the surface (i.e., temperature inversion). Less LCA also reduces downward LW radiation at the surface (FLDS), which leads to colder near-surface air than the observation, resulting in enhancement of the temperature inversion." (lines 276-279 in the revised manuscript)

5. **Rather than state that "most of the shaded area" is statistically significant, why not shade the statistically insignificant areas to avoid crowding the plot?**

→ In the revised manuscript, only the statistically significant areas were plotted.

Response to Reviewer #2

We sincerely appreciate Reviewer #2 for spending his/her invaluable time to give us lots of constructive, critical and helpful comments. Most comments were carefully reflected in the revised manuscript. Our responses to individual comments are listed below.

**Major comments**

1. **For the link between statement 3 and 4, the authors cite Park et al. (2014): "the horizontal and vertical transports of heat and moisture are the important factors inducing the net condensation of water vapor into cloud liquid (NCD) both in SAM0 and CAM5". Are these factors also important in the Arctic regions? How dominant are these factors? Are there any other important factors? Could other modifications in SAM0 contribute to the enhancement of NCD? For example, in-cloud turbulence and precipitation from super-cooled liquid clouds.**

   **The following analysis of concurrence and correlations between advection and clouds are not definite evidence either. As a model study (instead of observational data analysis), more concrete evidence is expected for this imperative link. How about adding a budget analysis like the one for statement 2 and 3 or experiments turning on/off certain model processes?**

   → As reviewer mentioned, we examined all of the cloud liquid and ice condensate process tendencies including in-cloud turbulences and precipitation in each grid-box of the model in the Arctic region (in figure 4 in the revised manuscript). By this effort, we found that net condensation of water vapor into cloud liquid (NCD) is a single dominant process for the Arctic cloud liquid condensation compared to other physical processes (e.g., nonlocal asymmetric turbulent eddies (CON), local symmetric turbulent eddies (PBL), precipitation (PRS), conversion of cloud liquid into cloud ice (WBF), etc.). Given the cloud fraction, in the model, the NCD is explicitly calculated by the saturation equilibrium in cloud macrophysics scheme, which indicates that the NCD in a grid-box is produced more with the condition of more water vapor and lower grid-mean temperature. Assuming that the Arctic region is a cylinder, the water vapor over the Arctic region can be increased by only the two ways, those are a convergence of meridional moisture flux and a surface moisture flux. Because the difference of surface moisture flux between the two models is much smaller than that of the convergence of meridional moisture flux in Arctic region (compare supplementary S3a with S3b in the revised manuscript), we infer that the differences in the large-scale horizontal advection of moisture and temperature from sub-Arctic to Arctic are responsible for the increase in the Arctic water vapor source. The Arctic temperature can be changed by various physical and dynamical processes such as radiation, grid-scale advection, moist turbulence, etc.. In this study, although SAM0 has a higher temperature than CAM5 in the Arctic, the net condensate rate in

SAM0 is larger than in CAM5 because of relatively larger poleward moisture transport into the Arctic. This point indicates that moisture transport is a dominant factor for the generation of NCD. We explained these points clearly on lines 199-206 and lines 237-241 in the revised manuscript.

Although we agree with the Reviewer #2, however, there is a difficulty to determine which factor indeed contributes to the increase in the NCD through the budget analysis, because NCD is calculated by a complex relationship between temperature, water vapor, cloud fraction in a grid-box (i.e., saturation equilibrium). Instead of additional budget analysis, we provide more evidence that SAM0 exhibits stronger convection (Supplementary S7) and explanations on how this can be related to the enhanced poleward moisture transport. (lines 230-236 in the revised manuscript)

2. **The authors speculate that horizontal advection rather than vertical transport is responsible for the enhancement of NCD due to the identical surface boundaries. Could the authors provide some discussion instead of speculation on the role of vertical transport? Is it possible that the modification of SAM0 on the convection scheme alters the vertical structure of the atmosphere, which could also contribute to the NCD enhancement?**

→ As the reviewer #2 commented, the replacement of convection scheme (i.e., UNICON scheme) in SAM0 can change the vertical structure of atmosphere as shown in Figure 5d and 5h in the revised manuscript. As a response to the comment, we examine the relative amount of annual-mean meridional moisture flux and vertical moisture flux averaged at 65° N (Figure R1). Figure R1 shows each flux difference between SAM0 and CAM5. For a fair comparison, we converted the vertical moisture flux unit to [g kg$^{-1}$ m$^{-1}$ s$^{-1}$]. The results indicate that the horizontal moisture flux is about 1000 times larger than vertical moisture flux.

[Figure]

**Figure R1. Differences the zonal-mean (a) meridional moisture flux and (b) vertical moisture flux at 65° N between SAM0 and CAM5**

**3. Evaluated against ERA-Interim, SAM0 exhibits smaller biases than CAM5 in annual-mean geopotential heights simulations. However, this does not guarantee a more accurate simulation of heat and moisture advection. For example, SAM0 overestimates geopotential height at 850 hPa in tropical Pacific and underestimates in sub-Arctic Pacific, leading to larger heat transport into the Arctic. The moisture transport is also affected by the location of pathways. Could the authors evaluate the heat and moisture advection directly against observations? More importantly, this is the only way to validate the claim of "proper heat and moisture transport is the key process in simulating Arctic climate". Otherwise, the evidence could only support a claim of "enhanced heat and moisture transport improve Arctic climate simulations" because the larger advection could be an overcompensation in order to increase liquid clouds.**

→ We agree with reviewer's comments. From the additional calculation, we found that the SAM0 reduces the bias of the poleward heat and moisture transports against observation compared to the CAM5. We calculated the biases of poleward heat and moisture transports in both CAM5 and SAM0 against ERA-Interim reanalysis (Supplementary S4 and S5 in the revised manuscript). CAM5 overestimates both moisture and heat fluxes over the midlatitude region against the observation but underestimates those on the periphery (around 70° N) of the Arctic circle. Although the positive bias over the midlatitude region still remains, SAM0 reduces the biases of CAM5 on the periphery (around 70° N) of the Arctic circle. (lines 219-222 in the revised manuscript) The horizontal pattern also shows that the enhanced poleward moisture transport in SAM0 is in agreement with the observation. Figure R2 shows the vertically integrated annual mean moisture flux and its convergence from ERA-Interim and from those differences between SAM0 and CAM5. The poleward moisture flux increases particularly in the North Atlantic and North American regions where are similar to the regions with large poleward moisture transport in ERA-Interim.

[Figure]

**Figure R2. Vertically-integrated annual-mean moisture flux in m g kg$^{-1}$ s$^{-1}$ (arrow) and its convergence (shaded) from (a) ERA-Interim and from (b) those differences between SAM0 and CAM5. Black contour denotes the Arctic circle (65° N).**

4. **Page 1 Line 23-26, "proper simulation of poleward heat and moisture transport is one key factor for simulating Arctic clouds" could not be wrong. However, drawing this conclusion from an uncertainty level of "association" hurts rather than lends credibility, especially in a model study. Would the authors provide more evidence to substantiate this claim? Or this manuscript could focus on the improvement of SAM0 and thoroughly evaluate all the causes.**

→ We agree with the Reviewer. Even though, in this revision, we tried to provide more pieces of evidence that poleward transports are tightly related with the increase of NCD and proper simulation of the poleward transports reduce the bias, our results are only from the comparison between specific two models. Obviously, the results cannot be generalized to other models. Accordingly, we tried to tone down the sentence and other concluding remarks in the revised manuscripts.

**Minor comments**

14. **Page 6 Figure 3, have the authors looked at the seasonal breakdown? In winter, could larger moisture transport lead to more NCD to cloud ice?**

→ We examined the seasonal cycles of poleward moisture transport and NCD for both cloud liquid and ice condensate rate (Figure R3). In wintertime, NCD for cloud ice (NCD-ice) in CAM5 is larger than that in SAM0, but NCD for cloud liquid (NCD-liq) in CAM5 is less than that in SAM0 throughout the year. In both models, the poleward moisture fluxes are the largest from summer to autumn and the associated NCD-liq averaged over the Arctic region is nearly identical to the poleward moisture transport. The seasonal variability of the difference of NCD-liq averaged over the Arctic region is almost identical with that of the grid-mean RH, which explains why the Arctic liquid cloud amount increases from May to September as shown in Figure 1 in the revised manuscript. These points were described on lines 247-253 in the revised manuscript.

[Figure]

**Figure R3. Annual cycles of zonal-mean poleward moisture transport (PMT65) at 65° N, net condensation rate of water vapor into cloud liquid (NCD-liq, center solid line), and net condensation rate of water vapor into cloud ice (NCD-ice, center dashed line) averaged over the Arctic area from SAM0 (red line) and CAM5 (blue line) and the differences of NCD-liq (black bars) and relative humidity (green bars)**

15. **Page 8 Figure 4, what is the percentage of these differences relative to the absolute values of CAM5. If it's too messy to plot on the figure, a description in the text would be fine too.**

→ The difference of poleward moisture (heat) transport between SAM0 and CAM5 is about 10% (15%) of its climatology, respectively. We added this information on lines 216-217 in the revised manuscript.

17. **Page 9 Figure 5 a and b, the correlation seems to weaken in the recent years. Any reasons?**

→ We couldn't find out the reasons. We think that the issue is an interesting subject in the future work.

20. **Page 10 Figure 6c, I am confused about the unit: $10^3$ K kg m$^{-1}$ s$^{-1}$. Could it be $10^3$ Km$^{-1}$ s$^{-1}$ ?**

→ Yes. $10^3$ K m$^{-1}$ s$^{-1}$ is correct. We corrected the unit in figure 8c in the revised manuscript.

21. **Most of the paragraphs lack a topic sentence to guide readers. For example, the topic sentence for the first paragraph in Section 3 could be "SAM0 reduces the negative biases of CAM5 in liquid cloud simulations".**

→ Thanks. We entirely corrected paragraphs in the revised manuscript.

**Other minor issues**

→ We sincerely thank for reviewer's kind comments. All of the minor issues are fully considered in the revised manuscripts. In addition, some other errors are corrected.

Response for Reviewer #3

We sincerely appreciate Reviewer 3 for spending his/her invaluable time to give us lots of constructive, critical and helpful comments. Most comments were carefully reflected in the revised manuscript. Our responses to individual comments are listed below.

**Major comments**

1. **Improvements of simulated Arctic clouds: according to Figure 1, although SAM0 cloud fraction is closer to observations, significant biases still persist, especially in winter. The improvement in PR90 is marginal. I suggest the authors to also compare the liquid and ice water path to the observations (e.g. Lenaerts et al. 2017), because they are also important for cloud radiative effects. For example, does the decrease in cloud ice mass (Figure 2c) make it closer to observations?**

   → As reviewer's suggestion, we added liquid water path (LWP) and ice water path (IWP) bias against from ERA-interim data (Supplementary S1 in the revised manuscript). Additionally, we added the vertical profile of cloud contents from ERA-interim data in the Supplementary S2 in the revised manuscript. We tried to use satellite observation data as shown in Lenaerts et al. (2017), but we could not find the data processing method, so I inevitably used ERA-interim data. Lenaerts et al. (2017) showed the LWP and IWP from CloudSat-Calipso climatology (C-C in their paper) and the difference between ERA-Interim and C-C. We could speculate the comparison between the model results and C-C, indirectly. In our result, CAM5 considerably underestimates the total cloud content with underestimating both cloud liquid and ice amount against ERA-interim data. In the SAM0, Although cloud ice condensate is underestimated as much as CAM5, cloud liquid condensate is simulated closed to observation, which reduces the overall bias of CAM5 in total cloud condensate. We mentioned the point on lines 161-165 in the revised manuscript.

2. **Relationship between meridional fluxes and increased cloud liquid:**
   **a) Is vertical advection included in the meridional transport? Heat and moisture transport into the Arctic doesn't just happen in the horizontal plane. In fact, eddies transport moisture along (moist) isentropes.**

   → We agree with the reviewer. To provide the best answer for your question, we examine the relative amount of the meridional moisture flux and vertical moisture flux averaged at 65° N (Figure R1 in response for reviewer#2). Figure R1 shows each flux difference between SAM0 and CAM5. For a fair comparison, we convert the vertical moisture flux unit to [g kg$^{-1}$ m$^{-1}$ s$^{-1}$]. The results indicate that the meridional moisture flux is about 1000 times larger than vertical moisture flux. So, we think the vertical moisture flux can be ignored in this study.

**b) Even though increased moisture flux and Arctic liquid cloud are correlated, it doesn't provide causation.**

→ We agree with the reviewer's comments. We think we do not provide sufficient explanation on the link between poleward moisture transport and the Arctic net condensation of water vapor into cloud liquid (NCD) in the original manuscript. Given the cloud fraction, in the model, the NCD is explicitly calculated by the saturation equilibrium in cloud macrophysics scheme, which indicates that the NCD in a grid-box is produced more with the condition of more water vapor and lower grid-mean temperature. Assuming that the Arctic region is a cylinder, the water vapor over the Arctic region can be increased by only the two ways, those are a convergence of meridional moisture flux and a surface moisture flux. Because the difference of surface moisture flux between the two models is much smaller than that of the convergence of meridional moisture flux in Arctic region (compare supplementary S3a with S3b in the revised manuscript), we infer that the differences in the large-scale horizontal advection of moisture and temperature from sub-Arctic to Arctic are responsible for the increase in the Arctic water vapor source. The Arctic temperature can be changed by various physical and dynamical processes such as radiation, grid-scale advection, moist turbulence, etc.. In this study, although SAM0 has a higher temperature than CAM5 in the Arctic, the net condensate rate in SAM0 is larger than in CAM5 because of relatively larger poleward moisture transport into the Arctic. This point indicates that moisture transport is a dominant factor for the generation of NCD. We explained this point clearly on lines 199-206 and lines 237-241 in the revised manuscript.

We agree that the correlation between poleward moisture transport and Arctic NCD do not explain definite causality. In the revised manuscript, in addition to the discussion prementioned above, we focused on to explain more clearly the link between poleward moisture transport and NCD with analyzing seasonal variabilities between the two models (lines 247-252 in the revised manuscript). Additionally, we provide more evidence that SAM0 exhibits stronger convection (Supplementary S7) and explanations on how this can be related to the enhanced poleward moisture transport. (lines 230-236 in the revised manuscript)

3. **The results shown here are from atmospheric only GCMs. Since the results depend on the atmospheric heat transport, ocean coupling could potentially alter the results. Have the authors looked at whether the changes in heat and moisture fluxes are still robust in coupled SMA0?**

→ Thank you for your constructive suggestion. We agree to need the additional experiment using a fully-coupled model owing to air-sea interaction. The factor may be more important in recent years when the Arctic has undergone rapid warming and Arctic sea ice has drastically

declined. Currently, we do not have the present-day full-coupled experiment data set of both CAM5 and SAM0. We will try this subject in future work (line 327 in the revised manuscript).

**Minor comments**

1. **Please hatch the maps to show their significance level, instead of saying "most shaded areas exceed a 95% significance level". Since not all areas are significant, it is useful to know where it is not significant.**

   → We agree with reviewer's comment. In the revised manuscript, only the statistically significant areas were plotted.

2. **P3 L7: What microphysics scheme does SAM0 use? Is it the same for CAM5? If not, it can introduce additional sensitivity.**

   → First of all, we are sorry for causing the confusion admitting that we didn't provide sufficient description on the models in the original manuscript. In this study, the only difference between SAM0 and CAM5 is the UNICON scheme, which is a unified convection scheme that replaces 1) CAM5's deep and shallow convection scheme (Park, 2014a, 2014b), and 2) convective detrainment process (Park et al., 2017). Not only cloud microphysics but also other features such as dynamic core, PBL, etc. are exactly the same for both models. We described this on lines 80-81 in the revised manuscript.

3. **P3 L23: Are different periods from 1979 to 2015 selected to compare with the corresponding observations (CALIPSO and CERES)? If so, it should be specified, since the simulation period is much longer than what the observations cover. If not, are the results sensitive to the mismatch in periods?**

   → Yes. The integration period of both CAM5 and SAM0 is 36 years from January 1979 to February 2015 and we use the climatology of the CALIPSO-GOCCP from June 2006 to November 2010 and the climatology of CERES-EBAF from March 2000 to February 2013. The period chosen is limited by the availability of each satellite observation data. Although the period among the model outputs and the satellite observations are different, we think that the data could be comparable when using each climatology. To verification of this point, we plotted same figures (Figure 1a, Figure 9d,e,f, and Figure 10d,e,f in the revised manuscript) using model data averaged with the same period to each observation data (Figure R4 and R5). The results show almost the same as the figures averaged with whole period.

[Figure]

Figure R4. Identical with Figure 1 in the original manuscript except for adding the results averaged from June 2006 to November 2010 (dashed lines) for each model

[Figure]

Figure 5. (a), (b), and (c) are identical with Figure 7d,e,f in the original manuscript and (d), (e), and (f) are identical with Figure 8d,e,f in the original manuscript except for using data averaged from March 2000 to February 2013 for each model

4. **P4 L14-15: The TCA bias also varies seasonally. SAM0's improvement is the most significant in summer, but less so in winter. Have the authors investigated the seasonal and spatial variability in poleward moisture transport? The seasonal cycle combined with spatial maps could shed light on why clouds are underestimated in the Arctic.**

→ We investigated the seasonal and spatial variability of poleward moisture transport and NCD for cloud liquid in figure 6 and supplementary S3a in the revised manuscript. In seasonal variability perspective, the poleward moisture transport in SAM0 is simulated more than CAM5 throughout the year (Figure 6 in the revised manuscript). In both models, the poleward moisture fluxes are the largest from summer to autumn and the associated NCD for cloud liquid averaged over the Arctic region is nearly identical to the poleward moisture transport. The seasonal variability of the difference of NCD for cloud liquid averaged over the Arctic region is almost identical with that of the grid-mean RH, which explains why the Arctic liquid cloud amount increases from May to September as shown in Figure 1 in the revised manuscript. These points were described on lines 247-253 in the revised manuscript. Supplementary S3a in the revised manuscript shows the vertically-integrated annual-mean moisture flux and its convergence from ERA-Interim and from those differences between SAM0 and CAM5. The poleward moisture flux increases particularly in the North Atlantic and North American regions where are identical with the regions with large poleward moisture transport in ERA-Interim.

5. **P6 Figure 3: Are these budgets closed? Net tendency profiles for liquid and ice from each model can be added to these figures.**

→ Actually, there was a simple mistake in calculating the NCD for the cloud liquid in Figure 3a in the original manuscript. Although a little bit more value of NCD for the cloud liquid is calculated, it did not affect the overall description for Figure 3 in the original manuscript. After we fixed it in the revised manuscript, the sum of all tendencies for both cloud liquid and ice is closed to zero because we plotted all tendencies for both cloud liquid and ice condensates. So we think that it is not necessary to add the zero line.

6. **P8 Figure 4: According to the moisture and heat flux convergence, we expect increased liquid condensation thus more liquid clouds at around 70N. Is this the case? For example, in winter the total cloud increase is quite spatially uniform over the Arctic Ocean (Fig 7b).**

→ The TCA increase in SAM0 during wintertime is relatively larger on the periphery of the Arctic circle (between 60° N and 70° N) than the Arctic inside region. The TAC increases particularly in the North Atlantic and North American regions where are identical with the

regions with the greatest increase in the poleward moisture transport as shown in supplementary S3a in the revised manuscript.

7. **P8 L22-P9 L2: We all know that correlation does not mean causation. By just showing correlation, the causality is not proven. For example, it is also possible that both changes are caused by a third factor that is not in the analysis.**

→ We agree with your comment. We described this point previously in major comment 2-(b).

8. **P8 L13: LCA was never spelled out in the paper.**

→ We are sorry for the confusion. LCA stands for the low-cloud fraction. The definition of LCA has been added on line 263 in the revised manuscript.

9. **P9 L14: What makes these two models the outliers? Are there any physical reasoning to say so? One should not just pick and choose the models or discard the end members because they do not agree with your hypothesis.**

→ We would like to use all available CMIP5 models including the necessary variables. Although bcc-csm1-1-m and MPI-ESM-LR models are outliers in the relationship between the meridional moisture transport and TCA and LCA, the models show the correlation between the meridional heat transport and LWP ration.

10. **P10 Figure 6: Are the widths of black lines in (a) and (b) represent the spread of observed poleward moisture transport? If not, these lines are very misleading and unnecessary.**

→ The widths of black lines in the original manuscript Figure 6 means not "the spread of observed poleward moisture transport" but "just values of CALIPSO cloud fraction". We removed the line and pointed out the "values of CALIPSO cloud fraction" at y-axis in Figure 8a and 8b in the revised manuscript.

11. **P10 L7: Upward LW at TOA includes both clear sky and cloud effects. Have the authors look at the cloud radiative effect differences between the two models? The negative bias in upward LW at TOA in the Atlantic sector seems to get worse in SAM0. Is it because in this region, SAM0 is producing too much clouds comparing to the observations?**

→ We checked the longwave cloud radiative forcing (LWCF) during the wintertime. LWCF in SAM0 simulates a positive bias against observation in the Atlantic sector, in contrast negative bias in the land and sea-ice region because of a temperature inversion. Indeed, SAM0 produces a little bit more clouds over the Atlantic sector compared with the observation, as shown in Figure 9c in the revised manuscript. Thus increase in the cloud amount over the open-sea region reduce the FLUT in SAM0, which enhanced the negative bias of FLUT in CAM5.

12. **P10 L11: Figure 7 shows TCA, but the argument the authors give here involves LCA. Is the LCA change the same as TCA?**

→ We plotted the TCA in Figure 7 in the original manuscript for consistency in the entire manuscript. The LCA change is almost the same as TCA.

13. **P11 L7: Remove "a" before "summertime biases"**

→ Thanks. We corrected in the revised manuscript.

14. **P13 Figure 8: Does panel d) suggest that CAM5 SW cloud forcing is too weak at the surface? This would lead to a warm bias over the Arctic ocean. But g) shows a cold bias, which means that the LW cloud forcing bias (not enough warming at surface) dominates the net forcing. It would be helpful to see both LWCF and SWCF to get a fuller picture. I suggest the authors to plot LWCF and SWCF for both TOA and surface, at least to include them in the supplementary material.**

→ We agree with reviewer's comment. During summertime, the surface temperature response is not explained by only SWCF. Rather the surface temperature is determined by surface net flux of SW and LW. So we plotted the net SW radiation at the surface (FSNS) and net LW radiation at the surface (FLNS) and the sum of FSNS and FLNS in the original manuscript Figure 9. The difference of the sum of FSNS and FLNS between SAM0 and CAM5 agrees with the difference of near-surface temperature ($T_{2m}$). We add the plots of SWCF and LWCF at TOA in the Supplementary S8 in the revised manuscript.

15. **P14 L11: The wording "SAM0 remedies these problems" is too strong, given that significant cloud biases still persist.**

→ We agree with reviewer. We corrected inappropriate sentences not to describe too strongly in the revised manuscript.

[revised manuscript text omitted]

---

## Referee Report (RR1)

I appreciate the authors' response and the revision. Many points are made clearer in the revision, and the paper has been improved. I have a few additional minor comments:

L163-164: ERA-I is not equivalent to observations. Labeling ERA-I as observations can be misleading (especially in figures). For example, ERA-I can still have large biases compared to CloudSat+CALIPSO or CERES-EBAF when it comes to downwelling shortwave surface fluxes in the Arctic (as large as ~30 W/m2). LWP biases can be as large as ~50 g/m2 in the Norwegian and Barents Sea. I suggest the authors to replace "observations" with "reanalysis" when it's referring to ERA-I.

L224: Same as above.

L275: I do not understand why the authors keep showing figures of TCA, but the discussion is only relevant for LCA. Mid and high clouds can also affect TOA radiative fluxes, if they are optically thick enough, though it does not seem to be the case here. There is not a single figure showing that LCA is the dominant cloud type in the Arctic in SAM0. I suggest the authors replace TCA figures with LCA in the manuscript.

L400: repetitive title in reference

Figure 2: I appreciate the authors' efforts to show ground-based observations from NSA. Since the absolute values are large for FSDS and FLDS, I suggest to plot the difference between observations and models instead of absolute values, so that the readers can see right away which months show the largest improvements.

---

## Author Response (AR2)

Response to Reviewer #2

We sincerely appreciate Reviewer #2 for spending his/her invaluable time to give us lots of constructive, critical and helpful comments. However, it seems that the reviewer reviewed our draft not based on our final response to the reviewer but based on Authors' Comment (AC) submitted at 30th September 2019. The reason why we think like this is because the figure numbers (e.g., Fig.R8, Fig. R7) mentioned in the reviewer's final comment was not in our final response to the reviewer but in the AC. Given this special situation, all of the reviewer's comments were treated and reflected in the revised manuscript with utmost care. Our responses to individual comments are listed below.

**Major comments**

**1. First of all, is the file "acp-2019-199-manuscript-version5.pdf" the final revised manuscript? It seems some of the figures are older than those in the Author's response, eg, Fig. 3 vs. Fig. R2 and Fig. 6 vs. Fig. R8.**

→ We guess "Fig. R2" and "Fig. R8" you mentioned are Fig. R1 and Fig. R2 below, respectively. In the previous revision stage, we included Fig. R1 below not in the manuscript but in the supplementary information. According to your comment, we now provide vertical profiles of cloud properties including ERA-Interim data (Fig. R1) as Fig. 3 in the manuscript (line 164-166). Fig. 6, however, is maintained in the manuscript. Here, we provide "NCD-ice" in Fig. R2 below to check whether the larger moisture transport leads to the more NCD for cloud ice, as you asked in the previous minor comment #14. The result shows that, in both models, the poleward moisture transport affects only NCD for cloud liquid (NCD-liq in Fig. R2). Therefore, in the course of the study, we focus on NCD for cloud liquid.

[Figure]

**Figure R1. Annual-mean vertical profiles of grid-mean (a) cloud condensate mass (cloud liquid + cloud**

ice), (b) cloud liquid mass, (c) cloud ice mass averaged over the Arctic area from ERA-Interim (ERA, black line), SAM0 (red lines), and CAM5 (blue lines), and (d) the difference of cloud fraction between SAM0 and CAM5.

[Figure]

**Figure R2. Annual cycles of zonal-mean poleward moisture transport (PMT65) at 65° N, net condensation rate of water vapor into cloud liquid (NCD-liq, center solid line), and net condensation rate of water vapor into cloud ice (NCD-ice, center dashed line) averaged over the Arctic area from SAM0 (red line) and CAM5 (blue line) and the differences of NCD-liq (black bars) and relative humidity (green bars)**

**2. It will also be helpful if the authors could cite the revised manuscript in the Author's response after specific comments (or point out where in the manuscript), instead of just saying "revised in the manuscript".**
→ We think that the reviewer reviewed our draft not based on our final "response to the reviewer" but based on "Authors' Comment", as mentioned above. We cited the manuscript in detail in final "response to reviewer". In this version, we also cite the manuscript in detail.

**3. To follow on my third major comment, as I worried, SAM0 reduces negative biases in poleward moisture transport yet introduces more positive biases (Fig. S5). It is not surprising that more water vapor leads to more cloud liquid condensation in the Arctic in summer. An alternative explanation for the improved cloud simulation could be that SAM0 overestimates moisture transport into the Arctic that enhances NCD and thus increases cloud liquids. Nevertheless, SAM0 still underestimates cloud fraction and liquid clouds, especially over ocean/ice (Fig. S1), indicating that enhanced moisture transports may not be a key factor for better simulations of Arctic clouds and definitely not a solution. In order to prove enhanced moisture transport is a concrete improvement, more thorough validation (eg, seasonality and pathways) against multiple datasets is needed.**

→ We totally agree with your comment. As you mentioned, our study has not demonstrated that the poleward moisture transport in SAM0 is improved compared to that of CAM5. Instead, in the revised manuscript, we demonstrated that "enhanced heat and moisture transport improve Arctic climate simulations". All of the relevant sentences are corrected in the revised manuscript (line 25-26, line 234, and line 350-351).

**Other comments:**

**1. In Fig. R7, why not compare SAM0 and CAM5 against ERA-Interim directly?**

→ We guess "Fig. R7" you mentioned denotes horizontal pathway of poleward moisture transport and its convergence. This figure was provided to your major comment #3. We no longer demonstrate that poleward moisture transport of SAM0 is improved compared to that of CAM5 as we mentioned above. Please understand omitting this figure.

**2. Minor comment 17, since one major argument of this manuscript is that the strong correlation between moisture transports and NCD suggests a causal link, my question is relevant and within the scope of this study. (Minor comment 17: Page 9 Figure 5 a and b, the correlation seems to weaken in the recent years. Any reasons?)**

→ First of all, we are sorry for missing your question. To provide the best answer for your question, we plotted the same figure except for during recent years (1998-2014) (Fig. R3). We found that the correlation is high (correlation coefficient 0.82) even during recent years as much as during the entire period in both two models. Nevertheless, we think that the reason why the correlation seems to weaken in recent years in Fig. 7a and b is that the NCD for cloud liquid increased since 2000, unlike PMT65. One of the reasons for the increased NCD for cloud liquid may be the decline of Arctic sea ice. When the sea ice is melted, the surface moisture flux increases, which in turn can cause an increase in the Arctic water vapor source. In both models, we found that the surface moisture flux has increased over the region where sea ice melted in recent years (Fig. R4). Related research on the impact of climate change on Arctic clouds will be the subject of our future work.

[Figure]

**Figure R3 Interannual time series of the vertically-integrated annual-mean poleward moisture flux at 65°N (PMT65, black line) and net condensation rate of water vapor into cloud liquid (NCD, red line) averaged over the Arctic area during recent years from (a) CAM5 and (b) SAM0**

[Figure]

**Figure R4 Surface moisture flux during former period (1979-1999) (left panel) and during later period (2000-2014) (right panel) and the difference between the two periods (center) from CAM5 (upper) and SAM0 (bottom)**

**Response for Reviewer #3**

We sincerely appreciate Reviewer 3 for spending his/her invaluable time to give us helpful comments. Most comments were carefully reflected in the revised manuscript. Our responses to individual comments are listed below.

**L163-164: ERA-I is not equivalent to observations. Labeling ERA-I as observations can be misleading (especially in figures). For example, ERA-I can still have large biases compared to CloudSat+CALIPSO or CERES-EBAF when it comes to downwelling shortwave surface fluxes in the Arctic (as large as ~30 W/m2). LWP biases can be as large as ~50 g/m2 in the Norwegian and Barents Sea. I suggest the authors to replace "observations" with "reanalysis" when it's referring to ERA-I.**
**L224: Same as above.**
→ Thank you for your kind comment. We totally agree with you and replace all of them in the revised manuscript (line 164, 165, 229, 245, and 295).

**L275: I do not understand why the authors keep showing figures of TCA, but the discussion is only relevant for LCA. Mid and high clouds can also affect TOA radiative fluxes, if they are optically thick enough, though it does not seem to be the case here. There is not a single figure showing that LCA is the dominant cloud type in the Arctic in SAM0. I suggest the authors replace TCA figures with LCA in the manuscript.**
→ We agree with your comment. We replaced TCA with LCA in Fig. 9 and the related sentences were corrected in the revised manuscript (line 290-296). However, previous studies have reported that, in summertime, Arctic mid and high clouds are optically thick enough to affect the TOA radiative flux at altitudes above 6 km (e.g. Lawson and Zuidema, 2009). Therefore we concluded to discuss not LCA but TCA for SWCF and LWCF during summertime. We maintained the TCA in Fig. 10 and replaced LCA with TCA in the description of the figure in the revised manuscript (line 307, 315, and 317).

**L400: repetitive title in reference**
→ Thank you. We corrected that in the revised manuscript.

**Figure 2: I appreciate the authors' efforts to show ground-based observations from NSA. Since the absolute values are large for FSDS and FLDS, I suggest to plot the difference between observations and models instead of absolute values, so that the readers can see right away which months show the largest improvements.**
→ Thank you. We corrected that in the revised manuscript.

**Reference**
Paul Lawson, R. and Zuidema, P.: Aircraft microphysical and surface-based radar observations of summertime arctic clouds, J. Atmos. Sci., 66(12), 3505–3529, doi:10.1175/2009JAS3177.1, 2009.

[revised manuscript text omitted]

---

## Author Response (AR3)

**Response for Editor**

We greatly appreciate the Editor for very helpful comments. All comments were incorporated into the revised manuscript. The responses to individual comments are listed below.

**1) Please explicitly note the years in the caption of figures that show a comparison between models and observations.**

→ Thank you for your kind comment. We added the years in the caption of Fig. 1, 2, 3, 9, S1, S5, S6, and S7.

**2) Page 7 (line 239): I don't think this statement is accurate because RH doesn't have a linear relationship with moisture and temperature. Please revise or remove.**

→ We agree with your comment. We deleted the discussion regarding RH and modified the sentence as "More poleward transport of moisture in SAM0 enhances the NCD for cloud liquid." in line 238-239 in the revised manuscript.

**3) Page 7 (line 263): change "those with tops" to "fractional coverage of cloud"**

→ Thanks. We corrected it in line 262 in the revised manuscript.

**4) Page 8 (line 289): Please clarify "two radiations". Longwave and shortwave radiative fluxes?**

→Following the comment, we clarified the sentence by saying that with "the net forcing of SW and LW radiation" in line 291 in the revised manuscript.

**5) Page 8 (line 295-297): I don't think this statement of causal relationship is supported here. The smaller snow depth and surface albedo can be due to other factors such as less snowfall, not necessarily the warmer temperature. Please revise.**

→Thank you very much for the comment. We agree with your comment. To address your comment, we corrected the sentence as "It is possible that the associated increase of $T_{2m}$ from CAM5 to SAM0 in the Arctic pole decreases snow depth, but other factors (e.g., less snowfall) may also be responsible for this decrease." in line 297-299 in the revised manuscript.

**6) Page 8 (line 299): change "charges" to "changes"**

→ Thanks. We corrected it in line 300 in the revised manuscript.

**7) Page 9 (line 327-330): This can be removed or needs to be elaborated.**

→ Thanks. We deleted the paragraph in the revised manuscript.

Additionally, we got the professional English editing service, which corrected some wordings and grammar, but did not change the context of the draft.

[revised manuscript text omitted]

**Page 8: [1] Deleted**                                    Author

**Page 8: [1] Deleted**                                    Author

**Page 8: [1] Deleted**                                    Author

**Page 8: [1] Deleted**                                    Author

**Page 8: [1] Deleted**                                    Author

**Page 8: [1] Deleted**                                    Author

**Page 8: [1] Deleted**                                    Author

**Page 8: [1] Deleted**                                    Author

**Page 8: [1] Deleted**                                    Author

**Page 8: [1] Deleted**                                    Author

**Page 8: [1] Deleted**                                    Author

**Page 8: [1] Deleted**                                    Author

**Page 8: [1] Deleted**                                    Author

**Page 8: [1] Deleted**                                    Author

**Page 8: [2] Deleted**                                    Author

**Page 8: [2] Deleted**          Author

**Page 8: [2] Deleted**          Author

**Page 8: [2] Deleted**          Author

**Page 8: [2] Deleted**          Author

**Page 8: [2] Deleted**          Author

**Page 8: [2] Deleted**          Author

**Page 8: [2] Deleted**          Author

**Page 8: [2] Deleted**          Author

**Page 8: [2] Deleted**          Author

**Page 8: [2] Deleted**          Author

**Page 8: [2] Deleted**          Author

**Page 8: [2] Deleted**          Author

**Page 8: [2] Deleted**          Author

**Page 8: [2] Deleted**          Author

**Page 8: [2] Deleted**          Author

| Page 8: [2] Deleted | Author |
|---|---|

| Page 8: [2] Deleted | Author |
|---|---|

| Page 8: [2] Deleted | Author |
|---|---|

| Page 8: [2] Deleted | Author |
|---|---|

| Page 8: [2] Deleted | Author |
|---|---|

| Page 8: [2] Deleted | Author |
|---|---|

| Page 8: [2] Deleted | Author |
|---|---|

| Page 8: [2] Deleted | Author |
|---|---|

| Page 8: [2] Deleted | Author |
|---|---|

| Page 8: [2] Deleted | Author |
|---|---|

| Page 8: [2] Deleted | Author |
|---|---|

| Page 8: [2] Deleted | Author |
|---|---|

| Page 8: [2] Deleted | Author |
|---|---|

| Page 8: [2] Deleted | Author |
|---|---|

| Page 8: [2] Deleted | Author |
|---|---|

Page 8: [2] Deleted                                    Author

Page 8: [2] Deleted                                    Author